



# RADIv1: a non-steady-state early diagenetic model for ocean sediments in Julia and MATLAB/GNU Octave

Olivier Sulpis[1,2], Matthew P. Humphreys[3], Monica M. Wilhelmus[4,5], Dustin Carroll[5,6], William M. Berelson[7], Dimitris Menemenlis[5], Jack J. Middelburg[1], Jess F. Adkins[8]

[1]Department of Earth Sciences, Utrecht University, Utrecht, The Netherlands
[2]Department of Earth and Planetary Sciences, McGill University, Montreal, Canada
[3]Department of Ocean Systems (OCS), NIOZ Royal Netherlands Institute for Sea Research, Texel, The Netherlands
[4]Center for Fluid Mechanics, Brown University, Providence, USA
[5]Jet Propulsion Laboratory, California Institute of Technology, Pasadena, USA
[6]Moss Landing Marine Laboratories, San José State University, Moss Landing, USA
[7]Department of Earth Sciences, University of Southern California, Los Angeles, USA
[8]Geological and Planetary Sciences, California Institute of Technology, Pasadena, USA

*Correspondence to*: Olivier Sulpis (o.j.t.sulpis@uu.nl)

**Abstract.** We introduce a time-dependent, one-dimensional model of early diagenesis that we term RADI, an acronym
accounting for the main processes included in the model: chemical Reactions, Advection, molecular and bio-Diffusion, and
bio-Irrigation. RADI is targeted for study of deep-sea sediments, in particular those containing calcium carbonates ($CaCO_3$).
RADI combines $CaCO_3$ dissolution driven by organic matter degradation with a diffusive boundary layer and integrates state-
of-the-art parameterizations of $CaCO_3$ dissolution kinetics in seawater, thus serving as a link between mechanistic surface-
reaction modelling and global-scale biogeochemical models. RADI also includes $CaCO_3$ precipitation, providing a continuum
between $CaCO_3$ dissolution and precipitation. RADI integrates components rather than individual chemical species for
accessibility and is straightforward to compare against measurements. RADI is the first diagenetic model implemented in Julia,
a high-performance programming language that is free and open source, and it is also available in MATLAB/GNU Octave.
Here, we first describe the scientific background behind RADI and its implementations. Then, we evaluate its performance in
three selected locations and explore other potential applications, such as the influence of tides and seasonality on early
diagenesis in the deep ocean. RADI is a powerful tool to study the time-transient and steady-state response of the sedimentary
system to environmental perturbation, such as deep-sea mining, deoxygenation or acidification events.





# 1 Introduction

The seafloor, which covers ~70% of the surface of the planet and modulates the transfer of materials and energy from the biosphere to the geosphere, remains for the vast majority unexplored. Today, this rich, poorly understood ecosystem is threatened locally by deep-sea mining activities (e.g. ploughing of the seabed), because it contains abundant valuable minerals and metals essential for the energy transition (Thompson et al., 2018). The deep ocean is also being perturbed globally by climate change, including seawater acidification caused by the uptake of ~10 billion tons of anthropogenic carbon dioxide ($CO_2$) into the ocean each year (Perez et al., 2018; Gruber et al., 2019), roughly a quarter of our total annual emissions (Friedlingstein et al., 2020). In this context, it is important to improve our understanding of the seafloor's response to environmental change.

Accumulation of sinking biogenic aggregates and lithogenic particles at the seafloor provides reactive material that regulates the chemical composition of sediment porewaters. Whereas biogenic particles typically sink through the water column at rates from a few meters to hundreds of meters per day (Riley et al., 2012), the same particles accumulate in sediments much more slowly, typically a few centimeters per thousand years (Jahnke, 1996). The residence time of solid particles in the top centimeter of sediments is therefore very long (a few hundred or thousand years) compared to their residence time in the water column (a few weeks). Additionally, while solutes are dispersed by advection in the water column, molecular diffusion dominates in porewaters, which is slower. The long residence time of reactive solid material in surface sediments, coupled with the slow diffusive transport of dissolved species, can lead to large gradients in chemical composition between sediment porewaters and the overlying seawater, inducing solute fluxes between the two (Hammond et al., 1996). Thus, the top few millimeters of the seafloor play a significant role in many major marine biogeochemical cycles.

The overall rate of biogeochemical reactions is determined by the slowest, 'rate-limiting' step, which can be (i) transport to or from the reaction site or (ii) the reaction kinetics of the particle at the mineral-water interface. At the seafloor, the rate-limiting step for many biogeochemical reactions is solute transport via molecular diffusion through the sediment porewaters or through the diffusive boundary layer (DBL). The DBL is a thin film of water extending up to a few millimeters above the sediment-water interface in which molecular diffusion is the dominant mode of solute transport. The presence of a DBL above the sediment-water interface (Fig. 1) has been reported by several investigators (Morse, 1974; Archer et al., 1989b; Gundersen and Jørgensen, 1990; Santschi et al., 1991; Glud et al., 1994) and its thickness depends on the composition and roughness of the substrate, as well as on the flow speed of the overlying seawater (Chriss and Caldwell, 1982; Dade, 1993; Røy et al., 2002; Han et al., 2018). Diffusive fluxes of solutes across the sediment-water interface are driven by concentration gradients between the overlying seawater and the sediment column being considered. If most of the concentration gradient for a given solute occurs within the porewaters, rather than within the DBL, then the diffusive flux of this solute is termed 'internal' or 'sediment-side controlled' (Boudreau and Guinasso, 1982). Conversely, if the majority of the concentration gradient for a given solute is within the DBL, the chemical flux across the sediment-water interface is termed 'external' or 'water-side transport-controlled'. In practice, the chemical exchange of most solutes is controlled by a combination of both regimes termed



'mixed-control', such as dissolved oxygen (Jørgensen and Revsbech, 1985; Hondzo, 1998), radon (Homoky et al., 2016; Cook et al., 2018), and the products of calcium carbonate dissolution (Sulpis et al., 2018; Boudreau et al., 2020), which have concentration gradients on both sides of the sediment-water interface. Despite the importance of the DBL in controlling diffusive fluxes across the sediment-water interface, DBLs are not explicitly included in most models that simulate early diagenesis in the deep ocean.

70            Multiple numerical models simulating early diagenesis have previously been published (Burdige and Gieskes, 1983; Rabouille and Gaillard, 1991; Boudreau, 1996b; Cappellen and Wang, 1996; Soetaert et al., 1996b; Archer et al., 2002; Munhoven, 2007; Couture et al., 2010; Yakushev et al., 2017; Hülse et al., 2018), each with its own assumptions and best area of application (Paraska et al., 2014). For instance, most existing models are limited to a steady state and are thus unable to predict the transient sediment response to time-dependent phenomena such as tides, seasonal change, ocean deoxygenation, or

acidification. Moreover, most of these models do not take the presence of a DBL into account, even though diffusion through the DBL may control the overall rate of many biogeochemical reactions. Finally, as the landscape of computing software and programming languages evolves and improves computing efficiency and code accessibility, it is important to leverage emerging developments to implement new biogeochemical models. Here, we describe a new sediment porewater model built upon earlier work termed RADI, an acronym accounting for the main processes included in the model that control the vertical

distribution of solutes and solids: chemical Reactions, Advection, molecular and bio-Diffusion, and bio-Irrigation. The novelty of RADI is that it combines organic matter degradation-driven $CaCO_3$ dissolution (Archer et al., 2002) with a diffusive boundary layer (Boudreau, 1996b) and integrates the state-of-the art parameterization of $CaCO_3$ dissolution kinetics in seawater (Dong et al., 2019; Naviaux et al., 2019a). RADI thus links mechanistic surface-reaction modelling to global-scale biogeochemical models (Carroll et al., 2020). By integrating components (e.g., total alkalinity) rather than individual chemical

species (e.g., carbonate and bicarbonate ions), RADI is easy to compare to observations. RADI is implemented in two popular scientific programming languages: Julia and MATLAB/GNU Octave. To our knowledge, this is the first diagenetic model implemented in Julia (https://julialang.org), a high-level, high-performance, and cross-platform programming language that is free and open source (Bezanson et al., 2017). Here, we first describe the scientific background behind RADI and its implementations. Then, we evaluate its performance in three selected locations and explore other potential applications, such

as the influence of tides and seasonality on early diagenesis in the deep ocean.

## 2 Model description

        In the following section, we describe how reactions, advection, diffusion, and irrigation are implemented in RADIv1. Model variables are *italicized* and their name as coded in the model are shown in `monospaced` font. Tables 1 and 2 include an inventory of model variables and parameters and a list of nomenclature for chemical species, respectively.






**Table 1. Nomenclature of model parameters and variables**

| Variable | Model notation | Description | Equation # |
|---|---|---|---|
| General | | | |
| $Z$ | z_max | Total height of the sediment column | |
| $dz$ | z_res | Depth resolution | |
| $z$ | depths | Array of modelled depths within the sediment | |
| $T$ | stoptime | Total simulation time | |
| $dt$ | interval | Time steps | |
| $t$ | timesteps | Array of modelled timepoints | |
| $\varphi_z$ | phi | Porewater porosity | 1 |
| $\beta$ | phiBeta | Porosity attenuation coefficient | 1 |
| $\varphi_{s,z}$ | phiS | Solid volume fraction | 2 |
| $\theta^2$ | tort2 | Squared tortuosity | 20 |
| $F_v$ | Fvar | Solid deposition flux | |
| $v_w$ | var_w | Bottom waters solute concentration | |
| $\delta$ | dbl | Diffusive boundary layer thickness | |
| $T_w$ | T | Temperature | |
| Advection | | | |
| $x_0$ | x0 | Bulk burial velocity at the sediment-water interface | 11 |
| $x_\infty$ | xinf | Bulk burial velocity at "infinite" depth | 12 |
| $u$ | u | Porewater burial velocity | 13 |
| $w$ | w | Solid burial velocity | 14 |
| $Pe_{h,z}$ | Peh | One-half of the cell Peclet number | 18 |
| $\sigma_z$ | sigma | Number from Fiadero and Veronis (1977) | 17 |
| Reactions | | | |
| $c/p$ | RC | "Redfield" ratio for carbon | |
| $n/p$ | RN | "Redfield" ratio for nitrogen | |
| $p/p$ | RP | "Redfield" ratio for phosphorus | |
| $K_v$ | Kvar | Half-saturation constant for a given electron acceptor | |
| $K_v'$ | Kvari | Inhibition constant for a given electron acceptor | |
| $k_{reaction}$ | kvar | Rate constant for a given chemical reaction | |
| $f_{v,z}$ | fvar | Fractions of organic matter degraded by a given oxidant | 5, 6 |
| $\eta_{diss.\ ca.}$ | order_diss_ca | Reaction order for calcite dissolution | 8 |
| $\eta_{diss.\ ar.}$ | order_diss_ar | Reaction order for aragonite dissolution | 9 |
| $\eta_{prec.\ ca.}$ | order_prec_ca | Reaction order for calcite precipitation | 10 |
| $\Omega_{ca}$ | OmegaCa | Seawater saturation state with respect to calcite | 8, 10 |
| $\Omega_{ar}$ | OmegaAr | Seawater saturation state with respect to aragonite | 9 |
| Diffusion | | | |
| $d_z(v)$ | D_var_tort2 | Effective molecular diffusion coefficient | 19, 23 |
| $d_z°(v)$ | D_var | "Free-solution" molecular diffusion coefficient | 19 |
| $b_z$ | D_bio | Bioturbation coefficient | 21, 22 |
| $\lambda_b$ | lambda_b | Characteristic bioturbation depth | 22 |
| Irrigation | | | |
| $\alpha_z$ | alpha | Irrigation coefficient | 26, 27 |
| $\lambda_i$ | lambda_i | Characteristic depth for irrigation | 27 |





## 2.1 Model structure and fundamental equation

The set of diagenetic equations composing RADI are based on CANDI, the method-of-lines code by Boudreau (1996b). Unlike the model of Boudreau (1996b), RADI does not solve a set of reactive-transport differential equations but instead computes the concentrations of a set of solids and solutes at each time step following a time vector set by the user. For initial conditions, the user can choose between predefined uniform values (e.g., set all concentrations to zero) or a set of saved concentrations (e.g., from a previous simulation that has reached steady-state). $T$ is the total simulation time, d$t$ is the temporal

resolution, i.e., the interval between each timestep, and $t$ refers to the array of modelled timepoints. All time units are in years (a). The interface between the surface sediment and overlying seawater, conventionally set at a sediment depth $z = 0$, represents the top layer of RADI's vertical axis (Fig. 1). The bottom layer of the model is at a sediment depth $Z$. Between these limits, n layers are present, each being separated by a constant vertical gap d$z$. Depth units are in meters. The values assigned to d$z$ and d$t$ depend on the nature of the problem and on the kinetics of the chemical reactions. In the present study, all cases use d$z = 2$

mm and d$t = 1/128000$ a, i.e., ~4 minutes. If a lower d$z$ is used, d$t$ needs to be lowered as well to preserve numerical stability. In general, the ratio d$z$/d$t$ should be kept below a value of 256 m/a. If d$z$ is divided by two, d$t$ needs to be divided by two as well, and the speed at which RADI runs will be reduced by a factor of four.

**Table 2. Nomenclature of modelled chemical species.** All variables are concentrations, expressed in mol per m³ of solid for solid species and mol per m³ of water for solute species.

| Variable v | Model notation | Description |
|---|---|---|
| $[O_2]$ | dO2 | Dissolved oxygen |
| $[TAlk]$ | dalk | Total alkalinity |
| $[\Sigma CO_2]$ | dtCO2 | Dissolved inorganic carbon |
| $[Ca^{2+}]$ | dCa | Dissolved calcium |
| $[\Sigma NO_3]$ | dtNO3 | Dissolved inorganic nitrogen |
| $[\Sigma SO_4]$ | dtSO4 | Dissolved inorganic sulfate |
| $[\Sigma PO_4]$ | dtPO4 | Dissolved inorganic phosphorus |
| $[\Sigma NH_4]$ | dtNH4 | Dissolved inorganic nitrogen |
| $[\Sigma H_2S]$ | dtH2S | Dissolved inorganic sulfide |
| $[Fe^{2+}]$ | dFe | Dissolved iron |
| $[Mn^{2+}]$ | dMn | Dissolved manganese |
| $[POC_{refractory}]$ | proc | Refractory particulate organic carbon |
| $[POC_{slow}]$ | psoc | Slow-decay particulate organic carbon |
| $[POC_{fast}]$ | pfoc | Fast-decay particulate organic carbon |
| $[Calcite]$ | pcalcite | Calcite |
| $[Aragonite]$ | paragonite | Aragonite |
| $[MnO_2]$ | pMnO2 | Manganese (IV) oxide |
| $[Fe(OH)_3]$ | pFeOH3 | Iron (III) hydroxide |
| $[Clay]$ | pclay | Clay[1] |

[1]We consider all clay minerals to be montmorillonite (Al$_2$H$_2$O$_{12}$Si$_4$, molar mass = 360.31 g mol$^{-1}$)





RADI operates on a static, user-defined porosity profile. Porewater porosity, $\varphi_z$ in Fig. 1, refers to the porewater volume fraction in the sediment (dimensionless) and typically decreases exponentially with sediment depth due to steady-state

compaction. The porewater porosity profile is parametrized following Boudreau (1996b) as:

$$\varphi_z = \varphi_\infty + (\varphi_0 - \varphi_\infty)e^{-\beta z} \tag{1},$$

where $\varphi_\infty$ is the porewater porosity at great depth, $\varphi_0$ is the porewater porosity at the sediment-water interface, and $\beta$ is an attenuation coefficient expressed in m$^{-1}$. A typical deep-ocean porewater porosity profile is shown in Fig. 1. Here the measured porewater porosity profile at station 7 of cruise NBP98-2 (Sayles et al., 2001) is fit using $\varphi_\infty = 0.87$, $\varphi_0 = 0.915$, and $\beta = 33$

m$^{-1}$. The solid volume fraction ($\varphi_s$, dimensionless) is defined as:

$$\varphi_{s,z} = 1 - \varphi_z \tag{2},$$

and increases with sediment depth (as compaction forces squeeze porewaters out).

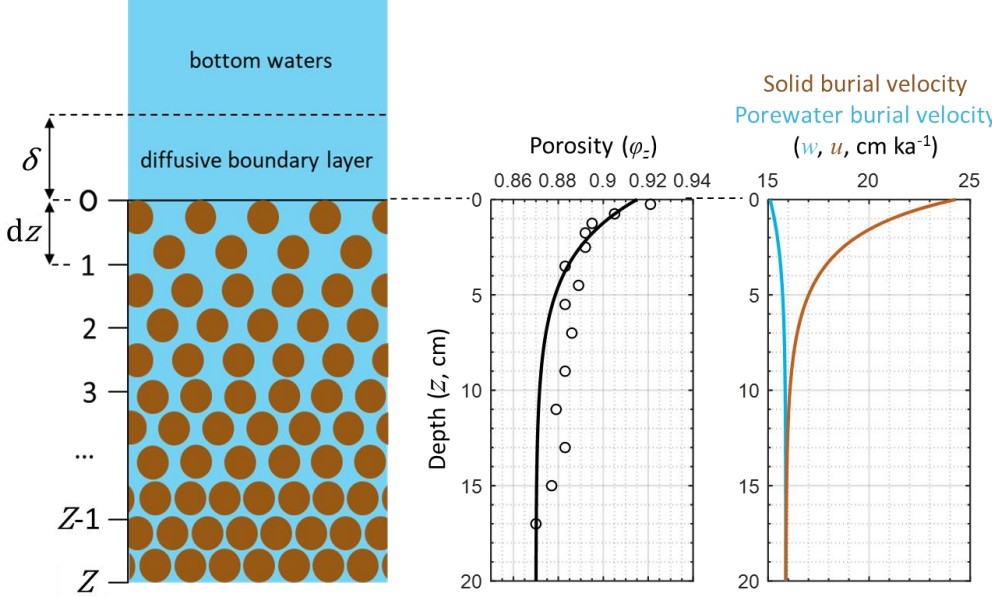

**Figure 1. Schematic of RADI's vertical structure alongside steady-state depth profiles of porewater porosity $\varphi_z$, see Eqs. (1,2), porewater ($u$, light blue solid line) and solid ($w$, brown solid line) burial velocities at in situ conditions taken at station 7 of Sayles et al. (2001). Burial velocity varies with depth due to porosity, as described in *Advection* Section 2.3. The open circles in the porewater porosity profile are porosity measurements from Sayles et al. (2001).**

Within this grid and for each time step, RADI computes the concentrations of 11 solute variables (TAlk, $\Sigma CO_2$, $O_2$, $Ca^{2+}$, $\Sigma NO_3$, $\Sigma SO_4$, $\Sigma PO_4$, $\Sigma NH_4$, $\Sigma H_2S$, $Fe^{2+}$, and $Mn^{2+}$) and 8 solid variables (*Calcite*, *Aragonite*, Fe(OH)$_3$, MnO$_2$, *Clay,* and three kinds of particulate organic carbon collectively termed POC). Concentration units are in mol per m$^3$ of water for solutes and in mol per m$^3$ of solid for solid species, For each modelled solute or solid concentration $v$ at time $t$ and sediment-depth $z$:





$$v_{(t+\mathrm{d}t),z} = v_{t,z} + \left[ R(v_{t,z}) + A(v_{t,z}) + D(v_{t,z}) + I(v_{t,z}) \right] \cdot dt \tag{3},$$

where $R(v_{t,z})$ quantifies the rate of change of $v_{t,z}$ due to chemical reactions, $A(v_{t,z})$ quantifies the rate of change of $v_{t,z}$ due to advection, $D(v_{t,z})$ quantifies the rate of change of $v_{t,z}$ due to molecular and bio-diffusion, and $I(v_{t,z})$ quantifies the rate of change of $v_{t,z}$ due to bio-irrigation. In general, only the subscript $z$s are explicitly written out in this document, for variables and parameters that vary with depth. The $t$s are implicit but excluded for clarity.



**Table 3. Diagenetic reactions, reaction rates, and reactions contributions to porewater TAlk and ΣCO₂.**

| Reaction | Rate [mM a$^{-1}$] | ΔTAlk | ΔΣCO₂ |
|---|---|---|---|
| **Organic matter degradation** | | | |
| $(CH_2O)(NH_3)_{\frac{n}{c}}(H_3PO_4)_{\frac{p}{c}} + O_2 \leftrightarrow CO_2 + \frac{n}{c} NH_3 + \frac{p}{c} H_3PO_4 + H_2O$ | $(k_{POC_{fast}}[POC_{fast}] + k_{POC_{slow}}[POC_{slow}])f_{O_2}$ | $+ n/c - p/c$ | + 1 |
| $(CH_2O)(NH_3)_{\frac{n}{c}}(H_3PO_4)_{\frac{p}{c}} + 0.8 NO_3^- \leftrightarrow 0.2 CO_2 + 0.4 N_2 + 0.8 HCO_3^- + \frac{n}{c} NH_3 + \frac{p}{c} H_3PO_4 + 0.6 H_2O$ | $(k_{POC_{fast}}[POC_{fast}] + k_{POC_{slow}}[POC_{slow}])f_{\Sigma NO_3}$ | $+ 0.8 + n/c - p/c$ | + 1 |
| $(CH_2O)(NH_3)_{\frac{n}{c}}(H_3PO_4)_{\frac{p}{c}} + 2 MnO_2 + 3 CO_2 + H_2O \leftrightarrow 4 HCO_3^- + 2 Mn^{2+} + \frac{n}{c} NH_3 + \frac{p}{c} H_3PO_4$ | $(k_{POC_{fast}}[POC_{fast}] + k_{POC_{slow}}[POC_{slow}])f_{MnO_2}$ | $+ 4 + n/c - p/c$ | + 1 |
| $(CH_2O)(NH_3)_{\frac{n}{c}}(H_3PO_4)_{\frac{p}{c}} + 4 Fe(OH)_3 + 7 CO_2 \leftrightarrow 8 HCO_3^- + 4 Fe^{2+} + \frac{n}{c} NH_3 + \frac{p}{c} H_3PO_4 + 3 H_2O$ | $(k_{POC_{fast}}[POC_{fast}] + k_{POC_{slow}}[POC_{slow}])f_{Fe(OH)_3}$ | $+ 8 + n/c - p/c$ | + 1 |
| $(CH_2O)(NH_3)_{\frac{n}{c}}(H_3PO_4)_{\frac{p}{c}} + 0.5 SO_4 \leftrightarrow HCO_3^- + 0.5 H_2S + \frac{n}{c} NH_3 + \frac{p}{c} H_3PO_4$ | $(k_{POC_{fast}}[POC_{fast}] + k_{POC_{slow}}[POC_{slow}])f_{\Sigma SO_4}$ | $+ 1 + n/c - p/c$ | + 1 |
| $(CH_2O)(NH_3)_{\frac{n}{c}}(H_3PO_4)_{\frac{p}{c}} \leftrightarrow 0.5 CO_2 + 0.5 CH_4 + \frac{n}{c} NH_3 + \frac{p}{c} H_3PO_4$ | $(k_{POC_{fast}}[POC_{fast}] + k_{POC_{slow}}[POC_{slow}])f_{CH_4}$ | $+ n/c - p/c$ | + 0.5 |
| **Redox reactions** | | | |
| $Fe^{2+} + 0.25 O_2 + 2 HCO_3^- + 0.5 H_2O \leftrightarrow Fe(OH)_3 + 2 CO_2$ | $k_{Fe\,ox}[Fe^{2+}][O_2]$ | - 2 | 0 |
| $Mn^{2+} + 0.5 O_2 + 2 HCO_3^- \leftrightarrow MnO_2 + 2 CO_2 + H_2O$ | $k_{Mn\,ox}[Mn^{2+}][O_2]$ | - 2 | 0 |
| $H_2S + 2 O_2 + 2 HCO_3^- \leftrightarrow SO_4^{2-} + 2 CO_2 + 2 H_2O$ | $k_{S\,ox}[\Sigma H_2S][O_2]$ | - 2 | 0 |
| $NH_3 + 2 O_2 + HCO_3^- \leftrightarrow NO_3^- + CO_2 + 2 H_2O$ | $k_{NH\,ox}[\Sigma NH_4][O_2]$ | - 2 | 0 |
| **CaCO₃ dissolution and precipitation** | | | |
| $CaCO_3 \leftrightarrow Ca^{2+} + CO_3^{2-}$ | $[Calcite] \cdot k_{diss.\,ca.} \cdot (1 - \Omega_{ca})^{\eta_{diss.\,ca.}}$ | + 2 | + 1 |
| $CaCO_3 \leftrightarrow Ca^{2+} + CO_3^{2-}$ | $[Aragonite] \cdot k_{diss.\,ar.} \cdot (1 - \Omega_{ar})^{\eta_{diss.\,ar.}}$ | + 2 | + 1 |
| $Ca^{2+} + CO_3^{2-} \leftrightarrow CaCO_3$ | $k_{prec.\,ca.} \cdot (\Omega_{ca} - 1)^{\eta_{prec.\,ca.}}$ | - 2 | - 1 |





**2.2 Reactions**

In RADI, biogeochemical reactions operate on solutes and solids throughout the entire sediment column, including
the very top and bottom layers. $R(v_z)$ is the net rate at which $v_z$ is being consumed (negative $R$) or produced (positive $R$) by
these reactions. Biogeochemical reactions in RADI (Table 3) are grouped into three categories: (i) organic matter degradation,
(ii) oxidation of reduced metabolites (organic matter degradation byproducts), and (iii) dissolution or precipitation of calcium
carbonate minerals. RADI has been designed for early diagenesis in deep-sea sediments, so formation and re-oxidation of
metal sulfide minerals are not considered.

**2.2.1 Organic matter degradation**

Organic carbon deposited on the seafloor originates mainly from primary production in the upper ocean or on land
and, to a lesser extent, from the ocean interior via chemoautotrophy. Despite the differences in origin, detrital organic matter
found in marine sediments typically has the same composition: ~60% proteins, ~20% lipids, ~20% carbohydrates, and a
fraction of other compounds (Hedges et al., 2002; Burdige, 2007; Middelburg, 2019). Here, the stoichiometry of organic matter
is represented by the coefficients $c$ (for carbon), $n$ (for nitrogen), and $p$ (for phosphorus). By default, the $c$:$p$ ratio is set to 106:1
and the $n$:$p$ ratio set to 16:1, following the Redfield ratio that describes the average composition of phytoplankton biomass
(Redfield, 1958), but these values can easily be adjusted. In RADI, $c/p$ is denoted RC, $n/p$ is denoted RN, and $p/p$ is denoted
RP, which is unity. Organic matter is also simplified here as an elementary carbohydrate ($CH_2O$). In reality, loss of H and O
during biosynthesis of proteins, lipids, and polysaccharides occurs (Anderson, 1995; Hedges et al., 2002; Middelburg, 2019),
which results in an effective molar ratio of $O_2$ consumed to C degraded of ~1.2 during aerobic respiration (Anderson and
Sarmiento, 1994) instead of 1 as assumed here (Table 3).

Observations show that some organic compounds are preferentially degraded and become selectively depleted (Cowie
and Hedges, 1994; Lee et al., 2000). As a result, the bulk reactivity of organic matter decreases with increasing age
(Middelburg, 1989). Degradation of organic matter deposited at the seafloor typically follows a sequential utilization of
available oxidants, $O_2$, $NO_3^-$, $MnO_2$, $Fe(OH)_3$, and $SO_4^{2-}$, followed by methanogenesis (Froelich et al., 1979; Berner, 1980;
Arndt et al., 2013). All organic matter degradation reactions implemented in RADI are shown in Table 3.

To account for the decrease in organic matter degradation rate with sediment depth, we separate organic matter into
fractions of different reactivity and we assign a rate constant to each of the degradable fractions. Following Jørgensen (1978),
Westrich and Berner (1984), and Soetaert et al. (1996b), three different classes of organic matter are considered: refractory,
slow-decay, and fast-decay. The refractory organic matter class is not reactive during the timescales considered here. The fast-
and slow-decay organic matter fractions each have a depth-dependent, oxidant-independent reactivity. The overall organic
matter degradation rate decreases with depth because the quantity of organic matter, the relative proportions of fast and slow-
decay materials, and the reactivities decline with depth. Organic matter is degraded following the sequential utilization of
available oxidants. The oxidant limitation is represented by a Michaelis-Menten-type (also termed 'Monod') function, in which





each oxidant has an associated half-saturation constant ($K_{oxidant}$ in mol m$^{-3}$) that symbolizes the oxidant concentration at which the process proceeds at half its maximal speed (Soetaert et al., 1996b). The presence of some oxidants may also inhibit other metabolic pathways; this is represented by an inhibition constant ($K_{oxidant}'$ in mol m$^{-3}$) that is specific to each oxidant. These limiting and inhibitory functions have been widely used (Boudreau, 1996b; Cappellen and Wang, 1996; Soetaert et al., 1996b; Couture et al., 2010) and they allow a single equation to be used for each component across the entire model depth range and

also permit some overlap between the different pathways, as observed in nature (Froelich et al., 1979). In RADI, the overall degradation of fast- or slow-decay organic carbon occurs at a rate:

$$R_{POC_{fast\ or\ slow},z} = f_{oxidant,z} \cdot k_{POC_{fast\ or\ slow},z} \cdot [POC_{fast\ or\ slow}]_z \qquad (4)$$

where $k_{POC,z}$ is the rate constant for the degradation of a given type of organic carbon (fast- or slow-decay) expressed in a$^{-1}$, $[POC_{fast\ or\ slow}]$ is the concentration of organic carbon (fast- or slow-decay) in sediments, and $f_{Ox.}$ is the sum of the fractions

of organic carbon degraded by each oxidant (dimensionless, always equal to one), given by:

$$f_{oxidant,z} = f_{O_2,z} + f_{\Sigma NO_3,z} + f_{MnO_2,z} + f_{Fe(OH)_3,z} + f_{\Sigma SO_4,z} + f_{CH_4,z} , \qquad (5)$$

where

$$f_{O_2,z} = \frac{[O_2]_z}{K_{O_2}+[O_2]_z} \qquad (6a),$$

$$f_{\Sigma NO_3,z} = \frac{[\Sigma NO_3]_z}{K_{\Sigma NO_3}+[\Sigma NO_3]_z}\frac{K_{O_2}'}{K_{O_2}'+[O_2]_z} \qquad (6b),$$

$$f_{MnO_2,z} = \frac{[MnO_2]_z}{K_{MnO_2}+[MnO_2]_z}\frac{K_{\Sigma NO_3}'}{K_{\Sigma NO_3}'+[\Sigma NO_3]_z}\frac{K_{O_2}'}{K_{O_2}'+[O_2]_z} \qquad (6c),$$

$$f_{Fe(OH)_3,z} = \frac{[Fe(OH)_3]_z}{K_{Fe(OH)_3}+[Fe(OH)_3]_z}\frac{K_{MnO_2}'}{K_{MnO_2}'+[MnO_2]_z}\frac{K_{\Sigma NO_3}'}{K_{\Sigma NO_3}'+[\Sigma NO_3]_z}\frac{K_{O_2}'}{K_{O_2}'+[O_2]_z} \qquad (6d),$$

$$f_{\Sigma SO_4,z} = \frac{[\Sigma SO_4]_z}{K_{\Sigma SO_4}+[\Sigma SO_4]_z}\frac{K_{Fe(OH)_3}'}{K_{Fe(OH)_3}'+[Fe(OH)_3]_z}\frac{K_{MnO_2}'}{K_{MnO_2}'+[MnO_2]_z}\frac{K_{\Sigma NO_3}'}{K_{\Sigma NO_3}'+[\Sigma NO_3]_z}\frac{K_{O_2}'}{K_{O_2}'+[O_2]_z} \qquad (6e),$$

$$f_{CH_4,z} = \frac{K_{\Sigma SO_4}'}{K_{\Sigma SO_4}'+[\Sigma SO_4]_z}\frac{K_{Fe(OH)_3}'}{K_{Fe(OH)_3}'+[Fe(OH)_3]_z}\frac{K_{MnO_2}'}{K_{MnO_2}'+[MnO_2]_z}\frac{K_{\Sigma NO_3}'}{K_{\Sigma NO_3}'+[\Sigma NO_3]_z}\frac{K_{O_2}'}{K_{O_2}'+[O_2]_z} \qquad (6f).$$

Half-saturation and inhibition constants for each oxidant used in RADI are given in Table 4. The degradation rate constant of

organic carbon, $k_{POC_{fast\ or\ slow}}$, is computed as a function of the flux of organic carbon reaching the seafloor and is sediment-depth dependent (Archer et al., 2002):

$$k_{POC_{fast},z} = (1.5 \times 10^{-1})\,(F_{POC} \cdot 10^2)^{0.85} \qquad (7a),$$
$$k_{POC_{slow},z} = (1.3 \times 10^{-4})\,(F_{POC} \cdot 10^2)^{0.85} \qquad (7b),$$

where $F_{POC}$ is the total flux of organic carbon reaching the seafloor (i.e., fast, slow, and refractory), in mol m$^{-2}$ a$^{-1}$. The numbers $1.3 \times 10^{-4}$ and $1.5 \times 10^{-1}$ have been tuned to best fit observations of both a Southern Ocean and North Atlantic Station, see the *Model evaluation* section 3.





### 2.2.2 Oxidation of organic-matter degradation byproducts

Organic matter degradation reactions primarily change oxidants (e.g., $O_2$, $NO_3^-$, $MnO_2$, $Fe(OH)_3$, $SO_4^{2-}$) into their
reduced forms (e.g., $H_2O$, $N_2$, $Mn^{2+}$, $Fe^{2+}$, $H_2S$; Table 3). If oxygen is introduced into the system, or if the reduced metabolites
diffuse upwards in oxic porewaters, then these reduced byproducts are converted back into their oxidized form and the energy
contained in them becomes available to the microbial community, though these energetics are not considered in RADI.

Here, four redox reactions involving organic-matter degradation byproducts are implemented (Table 3): oxidation of
each of $Fe^{2+}$, $Mn^{2+}$, $\Sigma H_2S$ and $\Sigma NH_3$. These four reactions consume porewater total alkalinity (TAlk) but do not alter porewater
$\Sigma CO_2$ (Table 3), thus locally acidifying porewaters. Here, we use the TAlk definition of Dickson (1981), in which TAlk is
defined as "*the number of moles of hydrogen ion equivalent to the excess of proton acceptors (bases formed from weak acids
with a dissociation constant $K \leq 10^{-4.5}$ and zero ionic strength) over proton donors (acids with $K > 10^{-4.5}$) in one kilogram of
sample*". This scheme should be sufficient for all open-ocean applications but may not be suitable for coastal and anoxic
environments with extensive metal-sulfide mineral turnover, which require a more complete set of redox reactions such as that
from the CANDI model of Boudreau (1996b). Additional components and reactions can easily be implemented in future
versions (see *Future Developments* section 5). The rate constants for these four redox reactions are taken from Boudreau
(1996b) and reported in Table 4.

**Table 4. Suggested values for model parameters.**

| Parameter | Model notation | Value | Unit | Source |
|---|---|---|---|---|
| $K_{O_2} / K_{O_2}'$ | KdO2/KdO2i | 3 / 10 | μM | Soetaert et al. (1996) |
| $K_{\Sigma NO_3} / K_{\Sigma NO_3}'$ | KdtNO3/KdtNO3i | 30 / 5 | μM | Soetaert et al. (1996) |
| $K_{MnO_2} = K_{MnO_2}'$ | KpMnO2/KpMnO2i | 42.4 | mM | Van Cappellen and Wang (1996)[1] |
| $K_{Fe(OH)_3} = K_{Fe(OH)_3}'$ | KpFeOH3/KpFeOH3i | 265 | mM | Van Cappellen and Wang (1996)[1] |
| $K_{\Sigma SO_4} = K_{\Sigma SO_4}'$ | KdtSO4/KdtSO4i | 1.6 | mM | Van Cappellen and Wang (1996)[1] |
| $k_{Fe\,ox}$ | kFeox | $10^6$ | mM$^{-1}$ a$^{-1}$ | Boudreau (1996b)[2] |
| $k_{Mn\,ox}$ | kMnox | $10^6$ | mM$^{-1}$ a$^{-1}$ | Boudreau (1996b)[2] |
| $k_{S\,ox}$ | kSox | $3 \times 10^5$ | mM$^{-1}$ a$^{-1}$ | Boudreau (1996b)[2] |
| $k_{NH\,ox}$ | kNHox | $10^4$ | mM$^{-1}$ a$^{-1}$ | Boudreau (1996b)[2] |
| $\beta$ | phiBeta | 33 | m$^{-1}$ | Tuned |
| $\lambda_b$ | lambda_b | 0.08 | m | Archer et al. (2002) |
| $\lambda_i$ | lambda_i | 0.08 | m | Archer et al. (2002) |

[1]Assuming a solid density of 2.65 g cm$^{-3}$; [2]Values for the "deep-sea"



### 2.2.3 CaCO₃ dissolution and precipitation

RADI includes two CaCO$_3$ polymorphs: low-Mg calcite and aragonite, but more could be added in future versions, e.g., high-Mg calcite and/or vaterite. Calcite and aragonite both have different dissolution kinetics, in which their dissolution rates increase as the undersaturation state of seawater with respect to calcite $(1 - \Omega_{ca,z})$ or aragonite $(1 - \Omega_{ar,z})$ increases (Keir, 1980; Walter and Morse, 1985; Sulpis et al., 2017; Dong et al., 2019; Naviaux et al., 2019b). Here, $\Omega_z$ is the sediment-depth-dependent saturation state of seawater with respect to calcite or aragonite, computed as $[Ca^{2+}]_z \cdot [CO_3^{2-}]_z / K^*_{sp}$, where $K^*_{sp}$ is the stoichiometric solubility constant of calcite or aragonite at in situ temperature, pressure, and salinity, as given in Mucci (1983) and Millero (1995). At each time step, $\Omega_z$ is computed using porewater $[Ca^{2+}]_z$ and $[CO_3^{2-}]_z$ from the previous time step, the latter being calculated as a function of TAlk and the proton concentration $[H^+]$. A Newton-Raphson method was used to calculate $[H^+]$ at each time step using TAlk, the fluoride, borate and silicate concentrations, $\Sigma PO_4$, $\Sigma SO_4$, $\Sigma NH_4$, and $\Sigma H_2S$. $[CO_3^{2-}]$ was then computed from the resulting $[H^+]$, $\Sigma CO_2$ from the previous time step and from the first and second carbonic acid dissociation constants at in-situ temperature and pressure of Lueker et al. (2000).

The dissolution rates ($R_{diss}$, in mol m$^{-3}$ a$^{-1}$) of calcite (solid blue line in Fig. 2a) and of aragonite (solid red line in Fig. 2a) as a function of $(1 - \Omega_{ca})$ are empirically defined as:

$$R_{diss.\ ca.,\ z} = [Calcite] \cdot k_{diss.\ ca.} \cdot (1 - \Omega_{ca})^{\eta_{diss.\ ca.}} \tag{8},$$

$$R_{diss.\ ar.,\ z} = [Aragonite] \cdot k_{diss.\ ar.} \cdot (1 - \Omega_{ar})^{\eta_{diss.\ ar.}} \tag{9}.$$

In these expressions, the dissolution rate constant ($k_{diss}$, in a$^{-1}$) and the reaction order ($\eta_{diss}$, unitless) are mineral-specific and implicitly account for each mineral's specific surface area. Similar formulations have previously been implemented to describe calcite dissolution rates (e.g. Archer et al. 2002) but, in most cases, with a high reaction order and with a tuned rate constant independent of solution chemistry (Fig. 2). Such discretizations are convenient but lack a mechanistic description of the controls on calcite dissolution in seawater (Adkins et al., 2021).

The latest advances using isotope-labelling approaches to study carbonate dissolution kinetics show abrupt changes in dissolution mechanism depending on solution saturation state with either calcite or aragonite (Subhas et al., 2017; Dong et al., 2019; Naviaux et al., 2019a; Naviaux et al., 2019b). Close to equilibrium, dissolution occurs primarily at sites on the crystal surfaces that are most exposed to the solution, e.g., steps and kinks. Further away from equilibrium, dissolution etch pits are activated at surface sites associated with defects and impurity atoms. Far away from equilibrium, there is enough free energy for dissolution etch pits to occur anywhere on the mineral surface, without the aid of crystal defects (Adkins et al., 2021). However, at temperatures most relevant to the deep oceans, ~5 °C or less, the defect-assisted dissolution mechanism is skipped (Naviaux et al., 2019b) and only the step-edge retreat (close to equilibrium) and homogeneous etch-pit formation (far away from equilibrium) dissolution regimes remain (Naviaux et al., 2019b) (Fig. 2). For both aragonite and calcite, while homogeneous etch-pit formation is indeed associated with a high-order dependency on the solution saturation state, step-edge retreat dissolution rates dominating near equilibrium show very little dependence on seawater saturation (Dong et al., 2019; Naviaux et al., 2019a). This could have significant consequences for the predicted carbonate dissolution rate near equilibrium:

saturation-state independent step-edge retreat dissolution will always be predicted to be faster close to equilibrium than dissolution associated with a high reaction order because a high reaction order forces the dissolution rate to converge to zero

as the solution gets closer to equilibrium (Fig. 2).

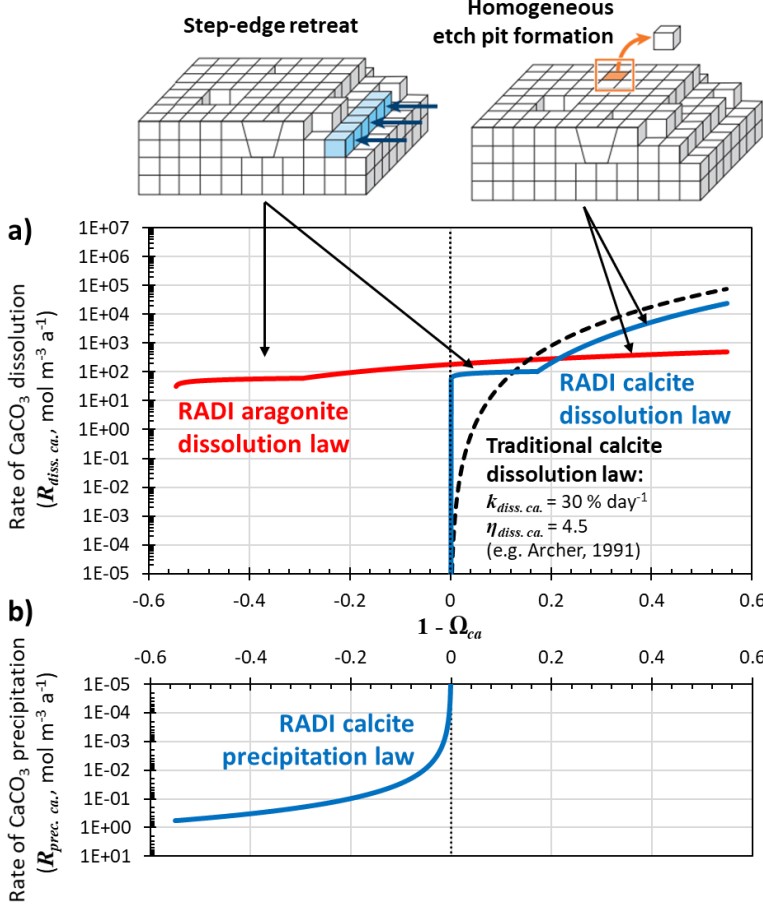

**Figure 2. (a) Dissolution rate of calcite as computed using Eq. (8) and [*Calcite*] = $10^4$ mol m$^{-3}$ (solid blue line), dissolution rate of aragonite as computed using Eq. (9) and [*Aragonite*] = $10^4$ mol m$^{-3}$ (solid red line), and a "classic" dissolution rate profile obtained**
**using [*Calcite*] = $10^4$ mol m$^{-3}$, a single dissolution rate constant for the entire (1-$\Omega_{ca}$) range $k_{diss}$ = 30 %$^{-d}$, and a reaction order $\eta_{diss}$ of 4.5 (dashed black line). (b) Precipitation rate of calcite as computed from Eq. (10). Note that dissolution rates are normalized here per total solid sediment volume, not per CaCO$_3$ surface area as in traditional kinetics studies.**

Naviaux et al. (2019a) derived reaction orders for two separate regions of the (1 - $\Omega_{ca}$) spectrum: the $\Omega_{ca}$ threshold

value dividing these two regions was $\Omega_{ca. critical} \approx 0.8$. Here, based on the results of Naviaux et al. (2019a), we set $\eta_{diss. ca.} = 0.11$

for $0.828 < \Omega_{ca} < 1$ and $\eta_{diss. ca.} = 4.7$ for $\Omega_{ca} \leq 0.828$. The $\Omega_{ca\ critical}$ value used here is slightly higher than the ~0.8 value given

in Naviaux et al. (2019a) in order to have a smooth transition between defect-assisted and homogeneous dissolution. For





aragonite, based on the results of Dong et al. (2019), we set $\eta_{diss.\ ar.} = 0.13$ for $0.835 < \Omega_{ar} < 1$ and $\eta_{diss.\ ar.} = 1.46$ for $\Omega_{ar} \leq 0.835$.

The rate constants are tuned to best fit the observations in the two stations presented in the *Model Evaluation* sections 3. We

use $k_{diss.\ ca.} = 6.3 \times 10^{-3}$ a$^{-1}$ for $0.828 < \Omega_{ca} < 1$, $k_{diss.\ ca.} = 20$ a$^{-1}$ for $\Omega_{ca} \leq 0.828$, $k_{diss.\ ar.} = 3.8 \times 10^{-3}$ a$^{-1}$ for $0.835 < \Omega_{ar} < 1$, and

$k_{diss.\ ar.} = 4.2 \times 10^{-2}$ a$^{-1}$ for $\Omega_{ar} \leq 0.835$. Both calcite and aragonite dissolution rate constants are lower than the values reported

in the original publications. We suspect that i) the reactive surface area of grains in sediments is much smaller than their

specific surface area measured using adsorption isotherms via the BET method and ii) unaccounted dissolution inhibitors are

present in sediments, such as dissolved organic carbon (Naviaux et al., 2019a). A comparison of the steady-state [$CO_3^{2-}$] and

[*Calcite*] porewater profiles predicted by RADI using the tuned rate constants implemented in RADIv1 and the original rate

constants is shown in Fig. S1.

Calcite precipitation is also included in the model and its rate (solid blue line in Fig. 2b) is parameterized with the

following function:

$$R_{prec.\ ca.,\ z} = k_{prec.\ ca.} \cdot (\Omega_{ca} - 1)^{\eta_{prec.\ ca.}} \tag{10},$$

where $k_{prec.\ ca}$ is the precipitation rate constant set to 0.4 mol m$^{-3}$ a$^{-1}$ and $\eta$ is equal to 1.76. The precipitation reaction order is

taken from Zuddas and Mucci (1998), corrected for a seawater-like ionic strength of 0.7 mol kg$^{-1}$. The precipitation / dissolution

rate continuum implemented in RADI (see Fig. 2) is very different from what a classic model with only calcite dissolution

following high-reaction order kinetics would display. For comparison, the dissolution rate of calcite using a dissolution rate

constant $k_{diss}$ of 30 % per day and a reaction order $\eta$ of 4.5, as implemented in most diagenetic models, including Archer

(1991), is shown in Fig. 2a. Mechanistic interpretations of the "kinks" in the dissolution rate profiles and of a non-zero

dissolution rate near equilibrium still require more research but the implications of these features for our understanding of

marine CaCO$_3$ cycles can be explored with the present model.

## 2.3 Advection

The bulk burial velocity at the sediment-water interface, $x_0$ in m a$^{-1}$, is given by:

$$x_0 = \left.\frac{\sum \frac{F_v \cdot M_v}{\rho_v}}{}\right/ \varphi_{s,0} \tag{11},$$

where $F_v$ is the flux of a solid species at the sediment-water interface (mol m$^{-2}$ a$^{-1}$), $M_v$ is the molar mass of that solid (g mol$^{-1}$), and $\rho_v$ is its solid density (g m$^{-3}$). The bulk burial velocity at greater depth, $x_\infty$, is computed as (Berner, 1980):

$$x_\infty = x_0\, \varphi_{s,0} / \varphi_{s,\infty} \tag{12}.$$

Thus, the porewater burial velocity, $u$, at all depths is:

$$u_z = u_\infty\, \varphi_{s,\infty} / \varphi_{s,z} \tag{13}$$

and the solid burial velocity, $w$:

$$w_z = w_\infty\, \varphi_{s,\infty} / \varphi_{s,z} \tag{14}.$$

Depth profiles of $u$ and $w$ are shown in Fig. 1, computed from the solid fluxes at station 7 of cruise NBP98-2 (Sayles et al., 2001), seen in *Model Evaluation* Section 3.2. In Fig. 1, the sharp porosity decline in the top centimeters of the sediments causes

the solid fraction at ~5-cm depth to be roughly 50% higher than just below the interface. This leads to a solid burial velocity decrease of about the same magnitude (Fig. 1).

Advection is implemented following Boudreau (1996b), where the advection rate ($A_z$, in mol m$^{-3}$ a$^{-1}$) for solutes is given by:

$$A_z(v) = -\left( u_z - \frac{d_z(v)}{\varphi_z} \cdot \frac{d\varphi_z}{dz} - d°(v) \cdot \frac{d\left( 1/\theta_z{}^2 \right)}{dz} \right) \cdot \frac{v_{(z+dz)} - v_{(z-dz)}}{2dz} \quad (15),$$

where $d_z$ is the effective diffusion coefficient for a given solute at a given depth (in m$^2$ a$^{-1}$), $d°$ is the "free-solution" molecular diffusion coefficient for a given solute (in m$^2$ a$^{-1}$), and $\theta_z$ is the depth-dependent tortuosity (unitless) defined in the *Diffusion section 2.4*. For solids, a more sophisticated weighted-difference scheme is employed (Fiadeiro and Veronis, 1977; Boudreau, 1996b):

$$A_z(v) = -\left( w_z - \frac{db_z}{dz} - \frac{b_z}{\varphi_{s,z}} \cdot \frac{d\varphi_{s,z}}{dz} \right) \cdot \frac{(1-\sigma_z)v_{(z+dz)} + 2\sigma_z v_z - (1+\sigma_z)v_{(z-dz)}}{2dz} \quad (16),$$

where $b_z$ is the depth-dependent bioturbation coefficient (m$^2$ a$^{-1}$) and

$$\sigma_z(v) = 1/\tanh(Pe_{h,z}) - 1/Pe_{h,z} \quad (17),$$

where

$$Pe_{h,z} = w_z \cdot dz/2b_z \quad (18).$$

The parameter $Pe_h$ is half of the 'cell Peclet number', which expresses the influence of advection relative to bioturbation across

a distance separating two points of the grid, centered at the depth $z$. If bioturbation dominates ($Pe_h << 1$), e.g., near the sediment-water interface, $\sigma_z$ tends toward zero and a centered-difference discretization is implemented. If advection dominates ($Pe_h >> 1$), e.g., deeper in sediments, $\sigma_z$ tends toward unity and backward-difference discretization prevails, see Eq. (16). This differencing scheme, originally developed by Fiadeiro and Veronis (1977), maintains stability in the entire sediment column (Boudreau, 1996b).

**2.4 Diffusion**

The diffusion flux of any species depends on its effective diffusion coefficient, $d_z(v)$, which varies with depth within the sediment.

For each solute, free-solution diffusion coefficients, denoted $d_z°(v)$, were computed at in-situ temperatures (Li and Gregory, 1974; Boudreau, 1997; Schulz, 2006). For solute variables representing several individual species (e.g., $\Sigma PO_4$, $\Sigma CO_2$),

the diffusion coefficient of the dominant species was considered. Given the high proportion of $HCO_3^-$ relative to $CO_3^{2-}$ and $CO_2{}_{(aq)}$ in seawater and porewaters, the diffusion coefficient of $HCO_3^-$ was adopted for both TAlk and $\Sigma CO_2$. Free-solution diffusion coefficients, their temperature dependencies and their sources are reported in Table 5. The diffusion of solutes in the



porewaters is slower than in an equivalent volume of water as a result of the physical barriers caused by the presence of solid grains in a sediment. To correct for this effect, we follow Boudreau (1996b) and compute the effective diffusion coefficient

for a given solute as:

$$d_z(v) = d°(v)/\left(\theta_z{}^2\right)$$ (19),

where so-called tortuosity ($\theta$) is defined as (Boudreau, 1996a):

$$\theta_z = \sqrt{1 - 2\ln(\varphi_z)}$$ (20).

**Table 5. Temperature-dependent molecular diffusion coefficients (m² a⁻¹).**

| Diffusion coefficient | Value | Source |
|---|---|---|
| $d_z°(TAlk)$ | $0.015179 + 0.000795 \times T_w$ | (Boudreau, 1997; Schulz, 2006)[1] |
| $d_z°(\Sigma CO_2)$ | $0.015179 + 0.000795 \times T_w$ | (Boudreau, 1997; Schulz, 2006)[1] |
| $d_z°(Ca^{2+})$ | $0.011771 + 0.000529 \times T_w$ | (Li and Gregory, 1974) |
| $d_z°(O_2)$ | $0.031558 + 0.001428 \times T_w$ | (Boudreau, 1997; Schulz, 2006) |
| $d_z°(\Sigma NO_3)$ | $0.030863 + 0.001153 \times T_w$ | (Li and Gregory, 1974)[2] |
| $d_z°(\Sigma SO_4)$ | $0.015779 + 0.000712 \times T_w$ | (Li and Gregory, 1974)[3] |
| $d_z°(\Sigma PO_4)$ | $0.009783 + 0.000513 \times T_w$ | (Boudreau, 1997; Schulz, 2006)[4] |
| $d_z°(\Sigma NH_4)$ | $0.030926 + 0.001225 \times T_w$ | (Li and Gregory, 1974)[5] |
| $d_z°(\Sigma H_2S)$ | $0.028938 + 0.001314 \times T_w$ | (Boudreau, 1997; Schulz, 2006) |
| $d_z°(Fe^{2+})$ | $0.001076 + 0.000466 \times T_w$ | (Li and Gregory, 1974) |
| $d_z°(Mn^{2+})$ | $0.009625 + 0.000481 \times T_w$ | (Li and Gregory, 1974) |

[1]value for $HCO_3^-$ ion, [2]value for $NO_3^-$ ion, [3]value for $SO_4^{2-}$ ion, [4]value for $HPO_4^{2-}$ ion, [5]value for $NH_4^+$ ion

For each solid, effective diffusion occurs through the mixing action of burrowing microorganisms, quantified using a bioturbation coefficient that decreases with depth. Archer et al. (2002) used a dataset including 53 sediment sites ranging in depth from 47 to 5668 m to derive an optimal bioturbation rate profile, in which the rate of bioturbation increases with increasing flux of total organic carbon reaching the seafloor ($F_{POC}$) and attenuates in low-oxygen conditions. This pattern was also observed by Smith et al. (1997) and Smith and Rabouille (2002). As in Archer et al. (2002), we couple both bioturbation

and irrigation to the incoming carbon deposition flux (Fig. 3) rather than water depth or sediment accumulation rate (Boudreau, 1994; Middelburg et al., 1997; Soetaert et al., 1996c), although all these quantities are related to each other. From an ecological perspective, more carbon to the seafloor represents more food available to benthic communities, hence more biological transport. Linking bioturbation activity to carbon deposition flux also allows for a direct coupling with Earth system models simulating carbon sinking fluxes in the ocean. Following Archer et al. (2002), we express the surficial bioturbation mixing

rate ($b_0$, in m² a⁻¹) as:





$$b_0 = (2.32 \cdot 10^{-6})(F_{POC} \cdot 10^2)^{0.85} \tag{21},$$

where $F_{POC}$ is expressed in mol m$^{-2}$ a$^{-1}$. The bioturbation mixing rate at all depths ($b_z$, in m$^2$ a$^{-1}$) is:

$$b_z = b_0 \, e^{-\left(z/\lambda_b\right)^2} \frac{[O_2]_w}{[O_2]_w + 0.02} \tag{22},$$

where the characteristic depth $\lambda_b = 8$ cm, following Archer et al. (2002), and $[O_2]_w$ is the oxygen concentration in the bottom

waters. This depth-dependent bioturbation mixing rate is common to all solids and its depth distribution is shown in Fig. 3 as

a function of in situ $[O_2]_w$ and $F_{POC}$. The effective diffusion coefficient for solids is then set as:

$$d_z(v) = b_z \tag{23}.$$

(Bio)diffusion is implemented in RADI following the centered difference discretization scheme from Boudreau

(1996b). At sediment depth $z$, where $0 < z < Z$, for both solutes and solids:

$$D_z(v) = d_z(v) \cdot \left(v_{(z-dz)} - 2v_z + v_{(z+dz)}\right)/(dz)^2 \tag{24},$$

where $d_z(v)$ is the relevant effective diffusion coefficient.

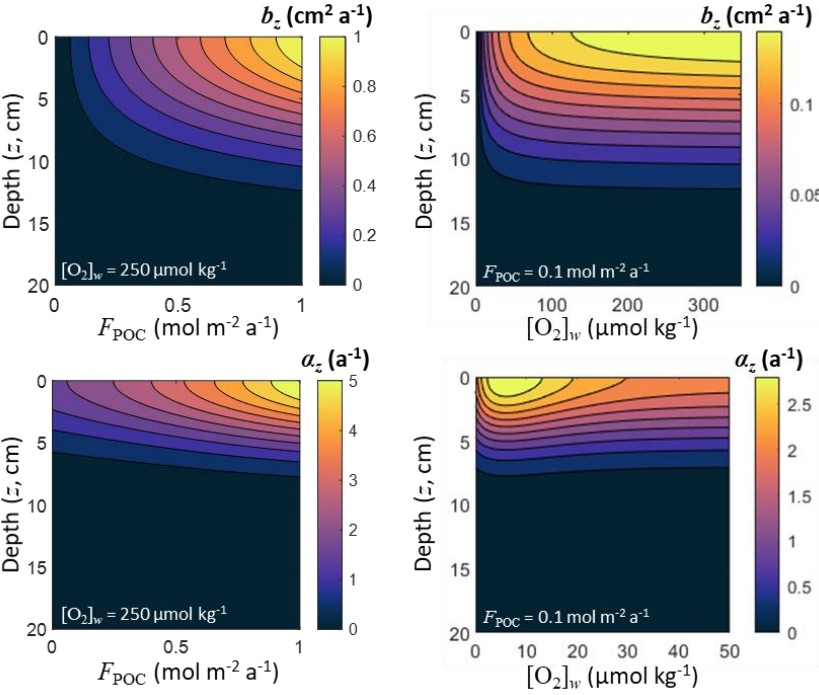

**Figure 3. Bioturbation mixing rate $b_z$ and irrigation coefficients $\alpha_z$ as a function of sediment depth $z$, organic carbon deposition**
**flux $F_{POC}$, and dissolved oxygen concentration in the bottom waters $[O_2]_w$.**





## 2.5 Irrigation

The mixing of solutes caused by burrow flushing or ventilation occurs through an ensemble of processes collectively termed *irrigation*. Macroscopic burrows are often present in the seafloor sediment, with a complex three-dimensional structure

and filled with oxygenated water that is ventilated for aerobic respiration. In a one-dimensional framework, this causes apparent internal sources or sinks of porewater solutes at particular depths (Boudreau, 1984; Emerson et al., 1984; Aller, 2001). Mathematically, this is parameterized as a non-local exchange function, i.e., a first-order kinetic reaction:

$$I_{t,z}(v) = \alpha_z (v_w - v_z) \tag{25},$$

where $\alpha_z$ is an irrigation coefficient common to all solutes expressed in a$^{-1}$. Following Archer et al. (2002), who used a dataset

of 53 sediment sites comprised of microelectrode oxygen profiles and chamber oxygen fluxes across the sediment-water interface to derive an irrigation-rate profile, we express the surficial irrigation coefficient as a function of the organic carbon deposition flux and the oxygen concentration of the overlaying waters:

$$\alpha_0 = 11 \left( \frac{\tan^{-1}\left(\frac{5F_{POC} \cdot 10^2 - 400}{400}\right)}{\pi} + 0.5 \right) - 0.9 + \frac{20[O_2]_w}{[O_2]_w + 0.01} \cdot \frac{F_{POC} \cdot 10^2}{F_{POC} \cdot 10^2 + 30} \cdot e^{\frac{-[O_2]_w}{0.01}} \tag{26}$$

and the irrigation coefficient at all depths as:

$$\alpha_z = \alpha_0 e^{-\left(\frac{z}{\lambda_i}\right)^2} \tag{27},$$

where the characteristic depth $\lambda_i$ is 5 cm (Archer et al., 2002). The depth-distribution of the irrigation coefficient is shown in Fig. 3 as a function of in-situ $[O_2]_w$ and $F_{POC}$.

## 2.6 Boundary conditions

Modeling of advection and diffusion processes requires appropriate boundary conditions in the layers above and

below ($z - dz$ and $z + dz$, respectively). Effective values of each variable immediately adjacent to the modelled depth domain are calculated following Boudreau (1996b) and used to compute the effects of advection and diffusion in the top and bottom layers using the same equations as within the sediment itself.

At the sediment-water interface, RADI enables prescribed solid fluxes and a diffusive boundary layer control for solutes. Following Boudreau (1996b), we calculate advection and diffusion at $z = 0$ for solutes and solids as:

$$v_{(-dz)} = v_{dz} + \frac{2\theta_z^2 dz}{\delta} (v_w - v_0) \tag{28}$$

and

$$v_{(-dz)} = v_{dz} + \frac{2dz}{b_0} \left( \frac{F_v}{\varphi_{s,0z}} - w_0 v_0 \right) \tag{29},$$

respectively. Here, $\theta$ is the tortuosity, $\delta$ is the boundary layer thickness (expressed in m, see Fig. 1), and $v_w$ is the solute concentration above the diffusive boundary layer, i.e., in the bottom waters. At the sediment depth $z = Z$, $v_{(Z+dz)}$ falls outside



the depth range of the model. The bottom boundary condition demands that concentration gradient disappear, which can be translated by the following for both solutes and solids:

$$v_{(Z+\mathrm{d}z)} = v_{(Z-\mathrm{d}z)} \tag{30}.$$

This 'no-flux' bottom boundary condition should be appropriate here because we set $Z$ so that all action occurs at shallower depth. However, if anaerobic methane oxidation or subsurface weathering are included in future versions, a 'constant' flux

boundary conditions might need to be included.

**2.7 Julia and MATLAB/GNU Octave implementations**

We have implemented RADI both in Julia (Humphreys and Sulpis, 2021) and in MATLAB/GNU Octave (Sulpis et al., 2021). Both implementations use similar nomenclature and provide identical results. Documentation for both is available from https://radi-model.github.io. The Julia implementation is available from https://github.com/RADI-model/Radi.jl and the

MATLAB/GNU Octave implementation is available from https://github.com/RADI-model/Radi.m.

Julia (https://julialang.org) is a high-level, high-performance, and cross-platform programming language that is free and open source (Bezanson et al., 2017). Its high-performance stems primarily from just-in-time (JIT) compilation of code before execution, which has been built-in since its origin. RADI uses Julia's multiple-dispatch paradigm, a core feature of the language, which improves the readability of the code and reduces the scope for errors. Specifically, each modelled component

of the sediment column is either a porewater solute or a solid. These components are initialized in the model as variables either of `Solute` or `Solid` type. Advection and diffusion are governed by different equations for porewater solutes than for solids but the same top-level functions (`advect!` and `diffuse!`) can be used within RADI to calculate the effects of these processes for both component types; the multiple-dispatch paradigm ensures that the correct equations are automatically used on the basis of the type of the input variable. While the model has been designed to solve a single profile at a time, Julia's

support for parallelized computation (across multiple processors) would also support efficient computations across a series or grid of vertical profiles.

As of version R2015b, MATLAB also features JIT compilation with a corresponding execution speed-up. However, MATLAB is an expensive, proprietary software, which limits how widely it can be used. The MATLAB implementation also runs in GNU Octave (https://www.gnu.org/software/octave/), which is a free and open-source clone of MATLAB. However,

GNU Octave executes more slowly than MATLAB for a variety of reasons, including a lack of JIT compilation.

For a model that necessarily includes long simulations with relatively short time-steps, computational speed is an important consideration. Our testing indicates that the Julia implementation runs ~3 times faster than the MATLAB (R2020a) implementation and ~70 times faster than the GNU-Octave implementation.

Simultaneously developing the model in two languages allowed us to quickly identify and remedy bugs and

typographical errors in both implementations. Each was coded independently, with equations and parameterizations written-out from the original sources, thus avoiding code copy-and-paste errors. Frequent comparisons were made throughout this process to ensure that the results were consistent. For a typographical error to survive to the final models would therefore





require an identical mistake to have been made independently in both implementations. The risk of such errors is thus substantially reduced by our dual-language approach. Where errors were identified, in some cases they were subtle enough
that they may otherwise not have been noticed, while still causing meaningful errors in final model results.

## 3 Model evaluation

To evaluate the performance of RADI, we used in-situ data obtained at three different locations and compared our predictions to the measured porewater and sediment solid phase composition profiles. We used these comparisons to tune the $CaCO_3$ dissolution and POC degradation rate constants; all other parameters were assigned a priori using values from the
literature. Thus we did not aim to reproduce observations as well as possible by tuning a wide selection of parameters. Instead, we evaluated whether a generic approach using measured deposition fluxes and bottom-water conditions could explain observations while tuning only the inorganic and organic reactivity constants.

### 3.1 North-western Atlantic Ocean

First, RADI was compared to the porewater and sediment composition measurements of station #9 described in Hales
et al. (1994), located in the North-western Atlantic Ocean (24.33°N / 70.35°W) at a 5210-m depth. The bottom-water TAlk and $\Sigma CO_2$ were 2342 µmol kg$^{-1}$ and 2186 µmol kg$^{-1}$, respectively, bottom-water in situ temperature was 2.2°C, salinity was 34.9, and oxygen concentration was 266.6 µmol kg$^{-1}$ (Hales et al., 1994). The computed bottom-water saturation state with respect to calcite was 0.88. The only $CaCO_3$ polymorph reaching the seafloor was assumed to be calcite. The calcite flux to the seafloor was set to 0.20 mol m$^{-2}$ a$^{-1}$ (20.02 g m$^{-2}$ a$^{-1}$) and the POC flux to 0.18 mol m$^{-2}$ a$^{-1}$, which correspond to the low end
of fluxes measured by sediment traps on the continental slope (Hales et al., 1994). The clay flux was set to a value of 26 g m$^{-2}$ a$^{-1}$ to fit the calcite sediment surface concentration measured by Hales et al. (1994). The porosity at the sediment-water interface was set to that measured by Sayles et al. (2001) in the Southern Pacific Ocean station, see Fig. 1. Following the diffusive boundary layer distribution from Sulpis et al. (2018), $\delta$ at the station location was set to 938 µm. This value represents an annual-mean estimate derived using a number of assumptions, e.g., considering the sediment-water interface to be a
horizontal surface and neglecting sediment roughness. A complete description of the environmental parameters for this North-Atlantic station, along with their sources, is available in Table S1.

RADI was run using the environmental conditions described above and the steady-state concentration profiles of $O_2$, $\Sigma NO_3$, calcite, and POC were compared with observations. Complete methods for solute and solids measurements are described in Hales et al. (1994). Briefly, porewater $O_2$ concentration was measured both in-situ using microelectrodes and on-
board (along with $\Sigma NO_3$) from the retrieved box core (Hales et al., 1994). The $O_2$, $\Sigma NO_3$, calcite, and POC profiles, along with the steady-state TAlk and $\Sigma CO_2$ profiles, were compared to those obtained from the MUDS model of Archer et al. (2002), computed from the same bottom-water chemistry and sediment accumulation rates.

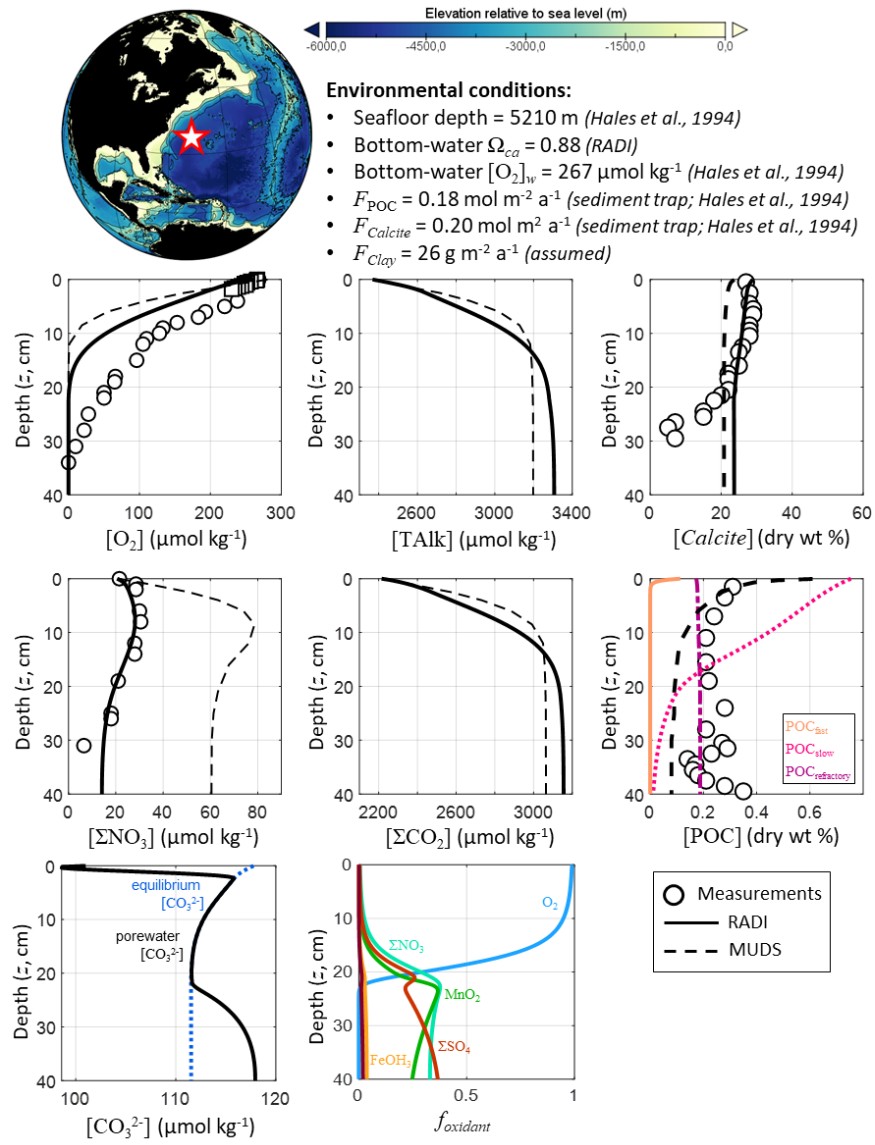

**Figure 4. Comparison of RADI with MUDS (Archer et al., 2002) and measurements from the North-western Atlantic station #9 described in (Hales et al., 1994). The lower panels represent (left) the computed $CO_3^{2-}$ concentrations in porewaters (solid black line) and at equilibrium with respect to calcite (dashed blue line), and (center) the fractions of organic matter degraded by a given oxidant.**

RADI predicts a porewater $O_2$ concentration decreasing from the bottom-water value to zero at ~20 cm-depth (Fig. 4). The MUDS model predicts a shallower oxygen penetration depth at ~15 cm-depth. In the top 2 cm, the RADI porewater $O_2$ predictions near the surface are in good agreement with the in-situ microelectrode measurements, whereas the MUDS-predicted $O_2$ profile appears steeper. Both the RADI and the MUDS predicted $[O_2]$ are lower than those measured on-board.

[$\Sigma NO_3$] is well reproduced by RADI and overestimated by MUDS. RADI predicts that organic matter respiration is mainly
aerobic (see Table 3a) until about 20 cm-depth. Between 20 and 35-cm depth, $\Sigma NO_3$ is the preferred oxidant for organic matter degradation (see Table 3b), which leads to a decrease in porewater [$\Sigma NO_3$] in both RADI predictions and observations, followed by $\Sigma SO_4$ deeper than 35-cm depth. The calcite profile is relatively well reproduced by both models in the top 20 cm but the measured calcite concentration drop below 20-cm depth is not reproduced by either model. Lower calcite concentrations below 20-cm depth may be attributed to a lower calcite accumulation rate to the seafloor in the past, whereas the models
consider accumulation of solids to be unchanged through time.

**3.2 Southern Pacific Ocean**

RADI was also compared with data collected at the station #7 mooring #3 described in Sayles et al. (2001), located in the Southern Pacific Ocean (60.15°S / 170.11°W), where the seafloor lies at the 3860-m depth. This dataset (http://usjgofs.whoi.edu/jg/dir/jgofs/southern/nbp98_2/) constrains the sedimentary system well: sediment-trap $CaCO_3$ and
POC fluxes, $CaCO_3$, and POC sediment-composition, sediment-porosity, and porewater solute depth profiles are all available from the same cruise and location.

The bottom-water chemical composition was taken from the GLODAPv2 1x1° climatologies (Lauvset et al., 2016), linearly interpolated over depth, latitude, and longitude to match the station location and seafloor depth. The bottom-water in-situ temperature was 0.84°C, salinity was 34.696, oxygen concentration was 215.7 μmol kg$^{-1}$, and calculated saturation state
with respect to calcite was 0.85. Solid fluxes at this station were measured by Sayles et al. (2001) using sediment traps collecting sinking particles between the months of November and December 1997. Their deepest sediment trap available was at a depth of 3257 m, i.e. 600 meters above the seafloor, which we assume to be representative of sinking fluxes to the seafloor, although the loss of material after collection usually causes sediment traps to underestimate the real sinking fluxes (Buesseler et al., 2007). The only $CaCO_3$ polymorph reaching the seafloor was assumed to be calcite and its flux was set to 0.25 mol m$^{-2}$
a$^{-1}$ (25.02 g m$^{-2}$ a$^{-1}$), rather than using the sediment-trap value, in order to fit the calcite sediment surface concentration measured by Sayles et al. (2001). This $CaCO_3$ flux to the seafloor is slightly higher than the measured $CaCO_3$ flux at 600 meters above the seafloor in mid-January 1997 (0.19 mol m$^{-2}$ a$^{-1}$; Sayles et al. 2001). The POC flux was arbitrarily set to 0.14 mol m$^{-2}$ a$^{-1}$ (4.62 g OM m$^{-2}$ a$^{-1}$), which is also slightly higher than the measured POC flux averaged between the months of November and December 600 meters above the seafloor (0.11 mol m$^{-2}$ a$^{-1}$). The clay flux, which we considered to be the total
measured sediment flux minus the assumed POC and calcite fluxes, was 32 g m$^{-2}$ a$^{-1}$. The porosity profile was tuned to best fit the porosity measurements at this station (Sayles et al., 2001, see Fig. 1). Finally, using the diffusive boundary layer world map computed in Sulpis et al. (2018) based on bottom current speeds at in-situ temperature and pressure measurements, the diffusive boundary layer thickness ($\delta$) at the station location was set to 715 μm. A complete description of the environmental parameters for this station, along with their sources, is available in Table S2.
RADI was run using the environmental conditions described above to compare the steady-state concentration profiles of TAlk, $\Sigma CO_2$, $O_2$, $\Sigma NO_3$, calcite, and POC to in-situ measurements. Methods for solutes and solids concentration analyses

are described in Sayles et al. (2001). Briefly, TAlk, $\Sigma CO_2$, and $\Sigma NO_3$ were sampled in situ using the Woods Hole Interstitial Marine Probe (Sayles, 1979) while $O_2$ was sampled at a higher depth-resolution but in the laboratory using whole-core squeezing (Bender et al., 1987). We also compare the RADI steady-state concentration profiles with those obtained from the

MUDS model described in Archer et al. (2002), computed with the same bottom-water chemistry and sediment accumulation rates.

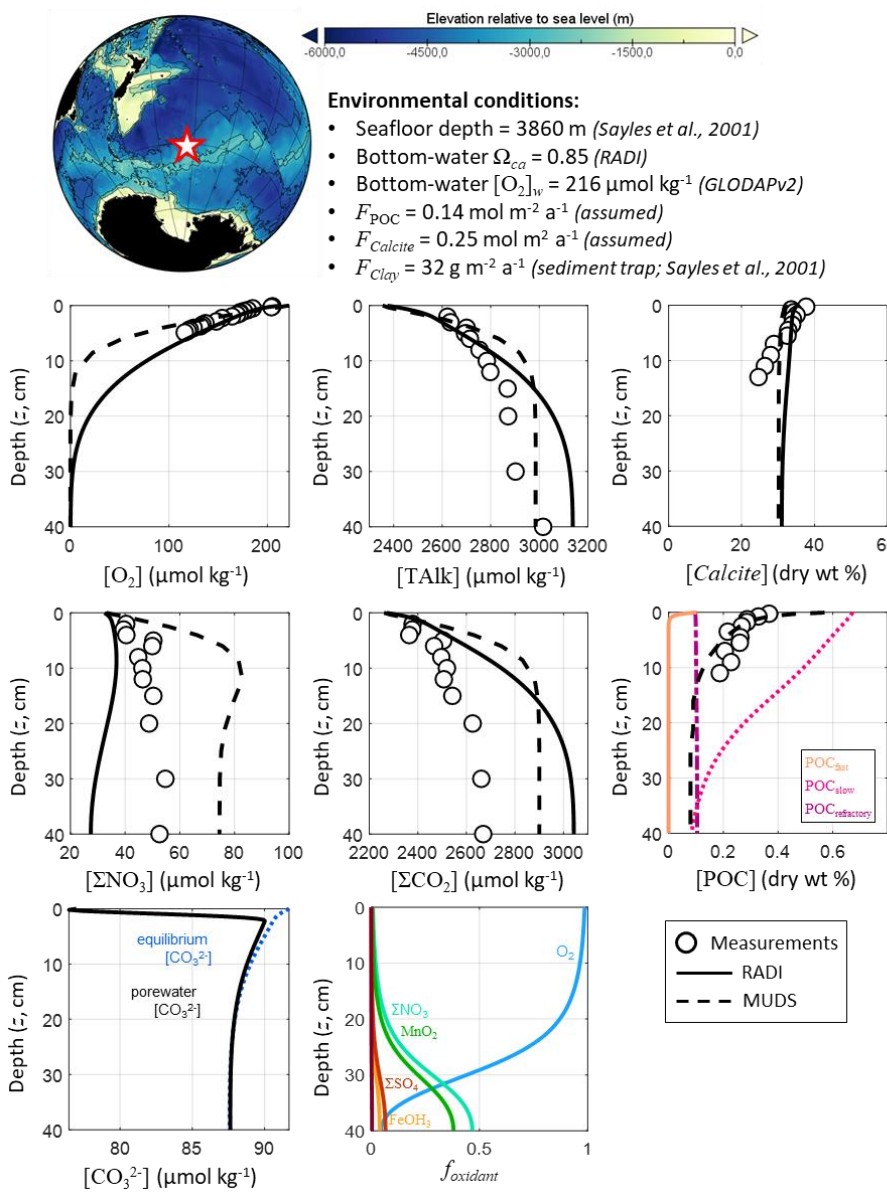

**Figure 5. Comparison of RADI with MUDS (Archer et al., 2002) and measurements from the station #7, mooring #3 MC-1 described in Sayles et al. (2001). The lower panels represent (left) the computed $CO_3^{2-}$ concentrations in porewaters (solid black line) and at equilibrium with respect to calcite (dashed blue line), and (center) the fractions of organic matter degraded by a given oxidant.**





While MUDS predicts porewater $O_2$ concentrations slightly below observations (**Fig. 5**), RADI predicts porewater $O_2$ concentrations that are slightly higher than observed. Because RADI does not predict porewater $O_2$ to go to zero until about the 30-cm depth, the dominant organic matter degradation pathway switches from mainly aerobic to $\Sigma NO_3$ at about the 30-cm

depth. Nevertheless, the RADI-predicted $\Sigma NO_3$ profile is lower than observed values, particularly toward the bottom of the resolved depth. The TAlk and $\Sigma CO_2$ porewater profiles predicted by MUDS and RADI are both larger than the observed, especially for $\Sigma CO_2$.

### 3.3 Central equatorial Pacific Ocean

As a third case study to evaluate the performance of RADI, solute fluxes through the sediment-water interface

computed from model steady-state runs were compared to fluxes measured using benthic chambers. The comparison took place at station #W-2 described in Berelson et al. (1994) and Hammond et al. (1996), located in the Central equatorial Pacific Ocean (0°N / 139.9°W) at a depth of 4370 m. Bottom-water in-situ temperature was set to 1.40°C and salinity to 34.69 (Lauvset et al., 2016). Bottom-water oxygen concentration was set to 159.7 μmol kg$^{-1}$ and the bottom-water saturation state with respect to calcite computed using the carbonate system solver within RADI was 0.78. For the purposes of this evaluation, the $CaCO_3$

flux to the seafloor was assumed to be entirely calcite. The calcite flux was set to 0.22 mol m$^{-2}$ a$^{-1}$, which represents 22.02 g of calcite m$^{-2}$ a$^{-1}$, the POC flux was 0.20 mol m$^{-2}$ a$^{-1}$, that is, 6.6 g of organic matter m$^{-2}$ a$^{-1}$, and the clay flux was set to 2.0 g m$^{-2}$ a$^{-1}$. The steady-state calcite content within the top cm was 61 dry wt %, in line with $CaCO_3$ contents observed in this area (Archer, 1996; Hammond et al., 1996). The porosity profile was built using an attenuation coefficient $\beta = 33$ m$^{-1}$, $\varphi_0 = 0.85$, which is the measured surface porosity (Hammond et al., 1996), and $\varphi_\infty = 0.74$, which is the measured porosity at depth

(Berelson et al., 1994), see Eq. (1). The DBL thickness, $\delta$, was fixed to an arbitrary value of 1 mm. A complete description of the environmental parameters for this station, along with their sources, is available in Table S3.

The diffusive fluxes for a given solute ($J_v$) between the sediment-water interface and the bottom waters occur as a response to the concentration gradient within the DBL and can be expressed by:

$$J_v = \varphi_0 D_v \times \frac{v_0 - v_w}{\delta} \tag{31},$$

where $v_0$ and $v_w$ are solute concentrations at the sediment-water interface and in bottom waters, respectively. In this definition, a positive $J_v$ indicates a solute release from the sediment porewaters to the bottom waters while a negative $J_v$ represents a solute flux towards the sediment.

The predicted TAlk, $\Sigma CO_2$, $\Sigma PO_4$, and $O_2$ fluxes (0.30 mol m$^{-2}$ a$^{-1}$, 0.32 mol m$^{-2}$ a$^{-1}$, 1.9 mmol m$^{-2}$ a$^{-1}$, and -0.23 mol m$^2$ a$^{-1}$, respectively) are all within the uncertainty bounds of the fluxes measured by benthic chambers at the same location

(0.28±0.09 mol m$^{-2}$ a$^{-1}$, 0.24±0.09 mol m$^{-2}$ a$^{-1}$, 1.4±0.5 mmol m$^{-2}$ a$^{-1}$, and -0.26±0.03 mol m$^{-2}$ a$^{-1}$, respectively, see Fig. 6). Nevertheless, the predicted $\Sigma NO_3$ flux (9.0 mmol m$^{-2}$ a$^{-1}$) is lower than its measured value (18±5 mmol m$^{-2}$ a$^{-1}$).



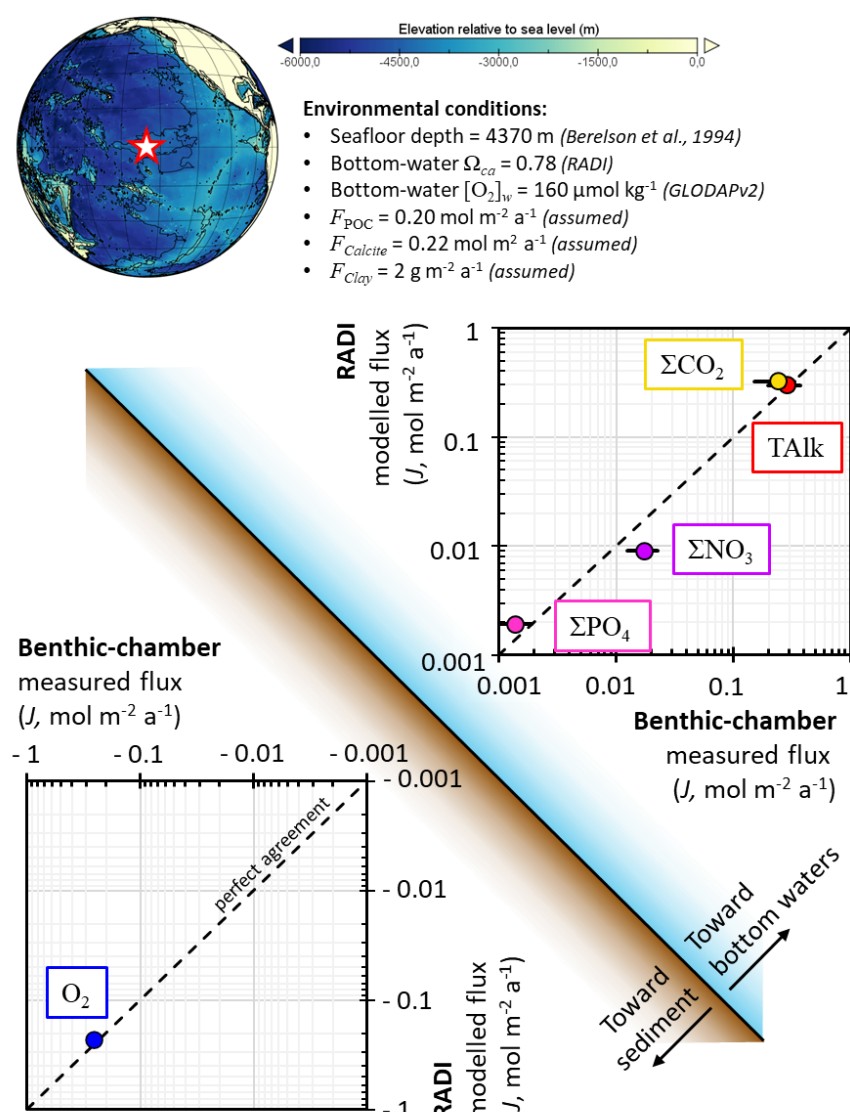

**Figure 6. Comparison of fluxes computed from RADI with benthic-chamber measurements from the station #W-2 described in Hammond et al. (1996). Error bars are included for all measured fluxes but not always visible.**


### 3.4 Discussion on model performance

These three model evaluation examples allowed us to determine a set of organic carbon degradation-rate constants, CaCO₃ dissolution-rate constants and organic-carbon flux composition (fast-decay, slow-decay or refractory) that can best reproduce sediment and porewater measurements in all stations while keeping all other model parameters to values from the

literature. In each station, bottom-water composition was fixed from observations. Due to a lack of adequate data, solid





deposition fluxes were tuned to best fit observed CaCO$_3$ and POC contents in sediments, except in the North-western Atlantic station where the CaCO$_3$ and POC fluxes were taken from measurements and in the Southern Pacific station where the clay flux was inferred from observations. The POC composition that allows to best fit porewater and sediment data in the three stations was as follows: 70% of fast-decay POC, 27% of slow-decay POC, and 3% of refractory POC. The tuned fast- and

slow-decay POC degradation rate constants are reported in *Organic matter degradation* section 2.2.1 and are of similar orders of magnitude as in most other models (Arndt et al., 2013). The tuned CaCO$_3$ dissolution rate constants are reported in CaCO$_3$ *dissolution and precipitation* section 2.2.3 and are two orders of magnitude lower than their laboratory-based values (Fig. S1), which we attribute to the presence of dissolution inhibitors (e.g., dissolved organic carbon, Naviaux et al., 2019a) or to the reactive surface area of natural grains in situ being lower than in laboratory experimental settings. Thus, with its current

settings, RADI should be able to accurately predict porewater chemistry and sediment composition in deep-sea environments, provided that the POC and CaCO$_3$ deposition fluxes are known.

Regarding porewater composition predictions, the largest difference between RADI and MUDS (Archer et al., 2002) seems to be with respect to nitrate dynamics. MUDS integrates a different organic-matter degradation scheme that includes organic-matter degradation rate constants specific to each oxidant. While RADI's $\Sigma NO_3$ follows observed porewater

concentrations in the North-western Atlantic, it strongly underestimates observations in the Southern Atlantic. Similarly, in the central equatorial Pacific, all RADI diffusive fluxes are within the uncertainty range of observations except the $\Sigma NO_3$ flux, which is underestimated by RADI. The low $\Sigma NO_3$ flux could be attributed, for example, to the presence of organic matter with a stoichiometry different than the Redfield ratio used in the current version of RADI or to errors in the nitrification parameters.

In addition, RADI and MUDS divergences regarding TAlk and $\Sigma CO_2$ porewater profiles may originate from their

different CaCO$_3$ dissolution kinetics (Fig. 2). On the one hand, MUDS includes classic high-order calcite-dissolution kinetics, which may explain that the MUDS TAlk and $\Sigma CO_2$ production is higher in the surface sediments than in RADI. On the other hand, RADI's step-edge retreat dissolution regime and its low reaction order induce calcite dissolution rates near equilibrium that are orders of magnitude higher than what is predicted in a high-order rate law such as that of MUDS (Fig. 2). That may cause RADI's higher TAlk and $\Sigma CO_2$ deeper in the sediment column, where porewaters get closer to equilibrium with respect

to calcite.

## 4 Potential model applications

In the following section, we continue to analyze the results obtained using the environmental conditions of the equatorial Pacific station #W-2 and present a few examples of relevant model applications. RADI can be used to study both steady state and transient conditions but in the following subsections we focus on time-dependent problems, since transient

diagenetic models are underrepresented in the literature.





## 4.1 Seasonal variability

At the seafloor, the fluxes of sinking material regulating the chemical composition of sediment porewaters are patchy in both space and time. Seafloor microbes and macrofauna respond quickly to pulses of organic matter delivery to the seafloor (Smith et al., 1992), causing short-term variability of sediment oxygen consumption (Smith and Baldwin, 1984; Smith et al., 665 1994). In addition, both the POC and $CaCO_3$ fluxes to the deep seafloor are strongly affected by seasonal flux variability (Billett et al., 1983; Smith and Baldwin, 1984; Lampitt, 1985; Lampitt et al., 1993; Lampitt et al., 2010). In the northeast Atlantic Ocean, at 3000-m depth, Lampitt et al. (2010) have shown that the summer POC and $CaCO_3$ fluxes can be ~ 10 and ~ 4 times higher, respectively, than the winter-time minima. The seasonal coupling between organic matter and $CaCO_3$ fluxes to the seafloor and the state of upper-ocean ecosystem is the result of rapid vertical transport of these materials (Sayles et al., 670 1994). If the fluxes of reactive material reaching the seafloor is affected by seasons, early diagenesis could display a seasonal signal and this should be taken into account when interpreting sedimentary data (Martin and Bender, 1988; Sayles et al., 1994; Soetaert et al., 1996a). Here we use RADI to assess how the porewater chemistry and solid composition of deep-sea sediments may be impacted by seasonally varying fluxes.

Seasonally time-varying solid fluxes to the seafloor ($F$, in mol m$^{-2}$ a$^{-1}$) can be represented with the following sinusoidal 675 function:

$$F(t) = F_{average} + \Delta F \sin\left(\frac{2\pi t}{\Delta t}\right) \tag{32},$$

where $t$ is time in years, $\Delta t$ is the time period separating two maxima (here set to 1 year), and $\Delta F$ is the amplitude. We assume that all $CaCO_3$ settling at the seafloor is calcite and set the mean $F_{Calcite}$ to 0.22 mol m$^{-2}$ a$^{-1}$, its amplitude change $\Delta F_{Calcite}$ to 0.11 mol m$^{-2}$ a$^{-1}$, the mean $F_{POC}$ to 0.20 mol m$^{-2}$ a$^{-1}$, and its amplitude change $\Delta F_{POC}$ to 0.10 mol m$^{-2}$ a$^{-1}$. All other parameters 680 correspond to values from the Central equatorial Pacific stations described in the *Model evaluation* section 3.3.

Seasonal variations in the calcite and POC fluxes reaching the seafloor are visible in sediment profiles of both solids and solutes (Fig. 7). Nonetheless, the amplitude of concentration changes separating an annual minimum from an annual maximum is very small, barely if (at all) detectable by observations. The annual amplitude is about 0.5 wt% for calcite, 0.07 wt% for POC, 2 μmol kg$^{-1}$ for [O$_2$], and 7 μmol kg$^{-1}$ for [ΣCO$_2$]. While concentrations at the sediment-water interface respond 685 quickly to seasonal flux changes, there is a phase lag that increases with depth between the concentrations of solids and porewater solutes and the seasonally changing fluxes to the seafloor. Thus, it is possible that porewaters and solid particles at the top mm-thick sediment layers are never really at a steady state but always lagging behind seasonal changes, even if these are minimal. This is in agreement with earlier modeling (Martin and Bender, 1988) and observational (Sayles et al., 1994) studies, indicating that biogeochemical reactions and bioturbation at the deep seafloor are too slow to show a discernible 690 seasonal signal. However, this might not be the case for sites receiving more reactive organic matter (Soetaert et al., 1996a).



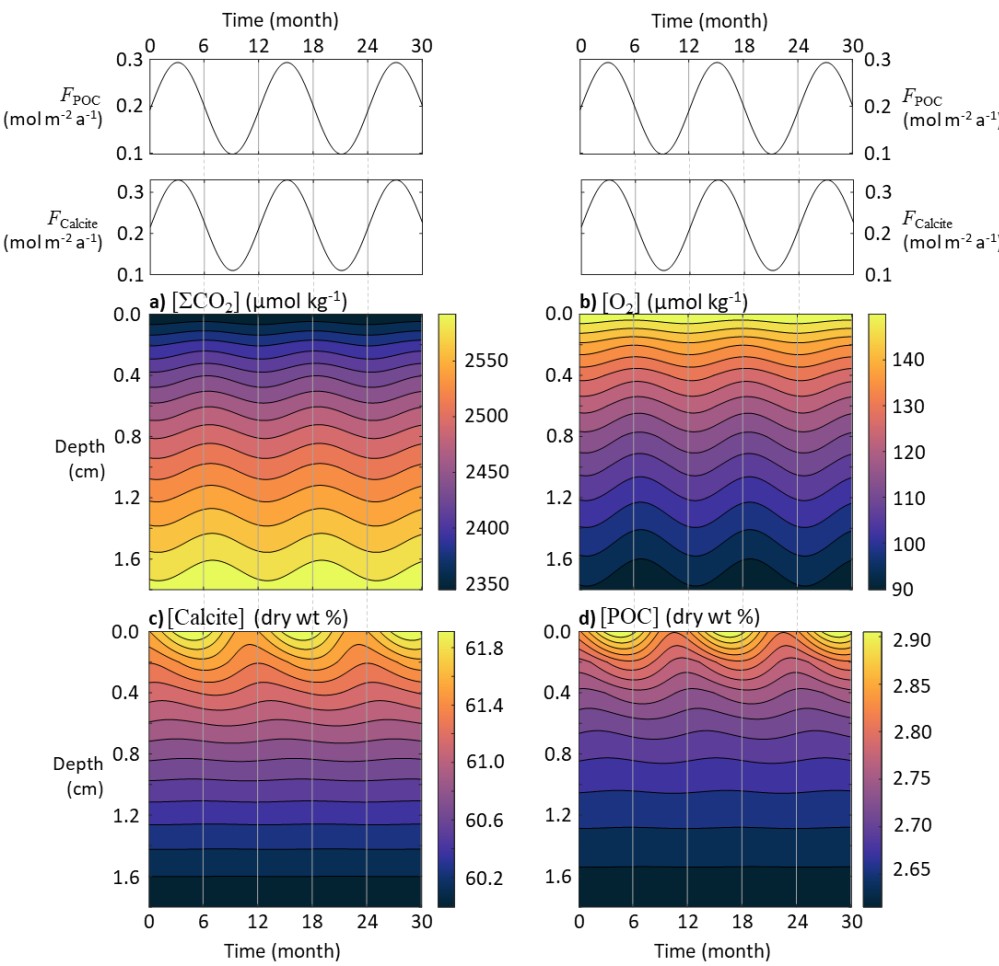

**Figure 7. Response of porewater $O_2$, $\Sigma CO_2$, calcite, and POC concentrations to seasonal fluctuations in the calcite and POC fluxes reaching the seafloor**

### 4.2 Tidal cycles

This section explores the applicability of RADI for studying the response of sediments to higher frequency phenomena such as tides. The DBL thickness is dependent on the overlying current speed (Levich, 1962; Santschi et al., 1983; Lorke et al., 2003): slower currents generate thicker DBLs whereas faster currents cause the DBL to thin (Larkum et al., 2003; Lorke et al., 2003; Higashino and Stefan, 2004). In the deep-sea, an important contributor of benthic current speeds are tidal forces, which makes tidal currents a potentially important contributor to biogeochemical exchanges across the sediment-water interface (Egbert and Erofeeva, 2002; Sulpis et al., 2019). If tidal current-speed fluctuations induce DBL thickness fluctuations, they may induce solute concentration fluctuations at the sediment-water interface, thus affecting early diagenesis. The strongest





tidal currents occur during the transition from high to low tides. For semidiurnal tides, the time period separating a low from a high tide is ~6 hours (Pugh, 1987). Setting the average DBL thickness to $\delta$ = 2mm and assuming that tides generate $\delta$

fluctuations with an amplitude $\Delta\delta$ = 1.5mm, the time-dependent $\delta$ can be expressed as:

$$\delta(t) = \delta_{average} + \Delta\delta \sin\left(\frac{2\pi t}{\Delta t}\right)$$ (33),

where $t$ is time in years, and $\Delta t$ is set to 1/1461 a ( ~6 hours). RADIv1 was run using the steady state solutes and solids depth profiles from the equatorial Pacific station #W-2 as initial conditions, with a DBL thickness fluctuating in response to tidal currents computed using Eq. (33).


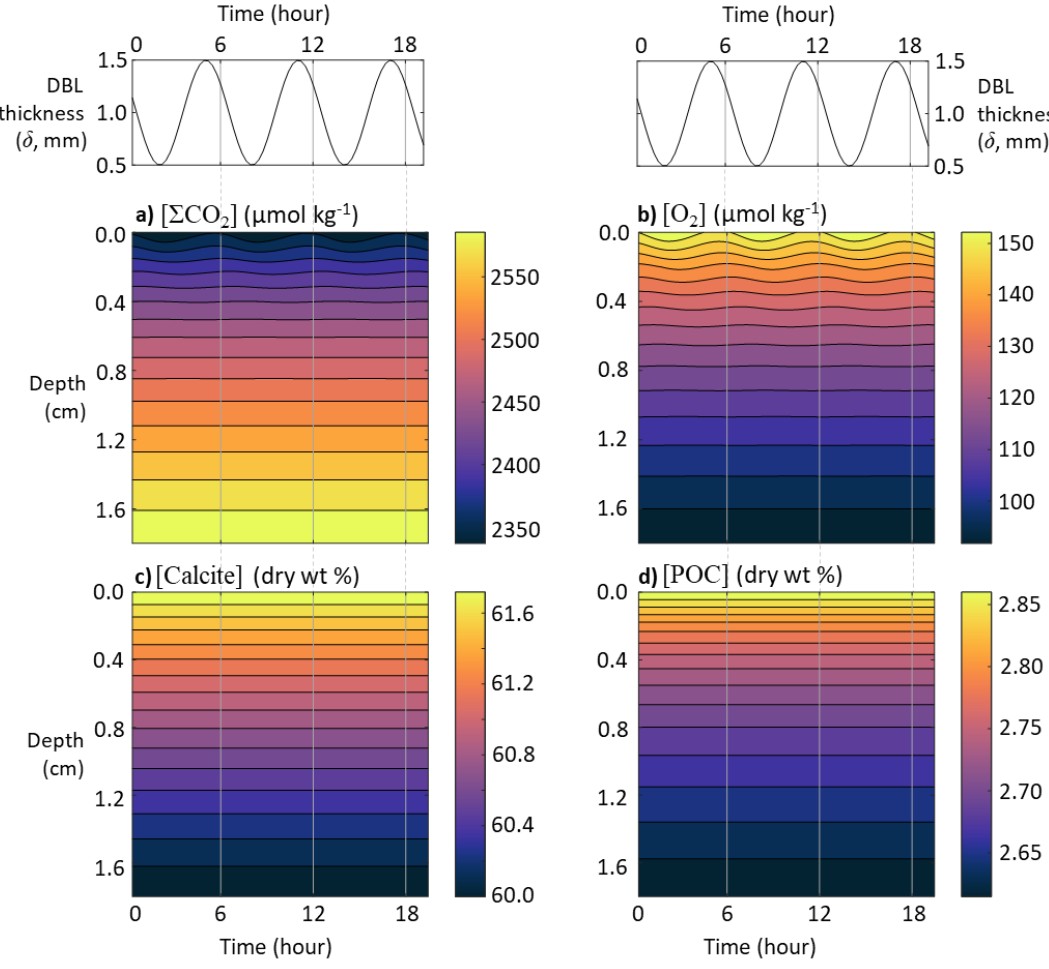

**Figure 8. Response of porewater O₂, ΣCO₂, calcite, and POC concentrations to fluctuations in DBL thickness ($\delta$) driven by tidal currents.**



While none of the solids seem to respond to tidal velocity fluctuation, due to their slow accumulation rate, solutes show a clear response (Fig. 8). At the sediment-water interface, the simulated porewater [$\Sigma CO_2$] variation amplitude within a single tidal cycle is about 25 µmol kg$^{-1}$ while [$O_2$] oscillates with an amplitude of about 8 µmol kg$^{-1}$. Deeper than a few millimeters below the sediment-water interface, the amplitude of both [$\Sigma CO_2$] and [$O_2$] changes become very small and tidal cycles are not measurable in the concentration profiles.

The implications are potentially important for our interpretation of porewater microprofiles. pH (Archer et al., 1989b; Cai and Reimers, 1993; Zhao and Cai, 1999; Cai et al., 2000), pCO$_2$ (Cai et al., 2000; Zhao and Cai, 1997), CO$_3^{2-}$ (de Beer et al., 2008; Han et al., 2014; Cai et al., 2016), O$_2$ (Revsbech et al., 1980; Reimers, 1987; Archer et al., 1989a; Sosna et al., 2007), and even dissolved Fe, Mn or S(-II) (Brendel and Luther, 1995) microelectrodes have been developed during the past decades. According to the results presented here, microprofiles, which capture instantaneous snapshots of porewater chemistry, should

show appreciable differences depending on when they are carried out during a tidal cycle. That organic matter degradation rates inferred from oxygen microprofiles span a wide range (Archer et al., 1989a; Arndt et al., 2013; Wenzhöfer et al., 2016) may be, among other factors, due to the dependency on tidal and other ocean-bottom current fluctuations. To adequately capture O$_2$ consumption rate in sediments, O$_2$ fluxes should be measured and integrated over a period of time longer than a tidal cycle.

### 4.3 Benthic chambers

RADI can also be used in the calibration of sensors and the optimization of sampling protocols and experimental designs. In the DBL, molecular diffusion is the dominant mode of solute transport and laboratory experiments of CaCO$_3$ dissolution in seawater suggest that diffusion through the DBL is the rate-limiting step for CaCO$_3$ dissolution at the seafloor in the absence of organic matter respiration (Sulpis et al., 2017; Boudreau et al., 2020). Nevertheless, earlier assessments of

in-situ CaCO$_3$ dissolution at the sediment-water interface in the central equatorial Pacific indicated that DBL thickness does not impact overall dissolution rates (Berelson et al., 1994).

In their study, Berelson et al. (1994) deployed a set of free-sinking benthic chambers onto the seafloor. In each chamber, the portion of the chamber exposed above the sediment-water interface was sealed and isolated from external bottom waters, and water samples were drawn during the incubation period. Each incubation lasted between 80 and 120 hours.

Chambers were stirred with a paddle at various rates to quantify the dependency of the measured diffusive fluxes across the sediment-water interface on the DBL thickness, which were calibrated via anhydrite dissolution to the 300-to-600 µm range. Seeing no influence of the stirring rate on the measured diffusive fluxes, Berelson et al. (1994) discarded the hypothesis of fast, surficial carbonate dissolution and instead argued for slow, high-reaction-order calcite dissolution kinetics at the seafloor, as subsequently implemented in most models. To better interpret the results from a benthic-chamber study such as that of

Berelson et al. (1994), RADI can be used to predict the time-response of diffusive fluxes across the DBL following an instantaneous change of DBL thickness due, for instance, to a change in paddle stirring rate within a chamber.





RADI was run using the steady-state solutes and solids depth profiles from the equatorial Pacific station #W-2 as
initial conditions. In the initial run, the DBL thickness was set to 1 mm. We simulated the response of this sediment to an
instantaneous change in the DBL thickness, with one model run representing a situation where $\delta$ increases from 1 to 5 mm
(e.g., a slow stirring rate) and one model run representing a $\delta$ drop from 1 mm to 200 μm (e.g., a fast stirring rate). Following
a 5-fold increase in $\delta$, diffusive fluxes of TAlk, $\Sigma CO_2$, and $O_2$ initially decrease by a factor ~5 but then increase back as the
solute concentrations at the interface adapt to the new DBL (Fig. 9). Two hours after the $\delta$ increase, diffusive fluxes converge
toward a new steady state. Following a $\delta$ 5-fold decrease, diffusive fluxes immediately increase by the same magnitude, but
go back close to their initial value within an hour as the interfacial porewater concentrations adjust to the new DBL. These
results suggest that the incubation periods of the Berelson et al. (1994) benthic chamber experiments were long enough to let
porewaters adjust to the changes caused by the paddle stirrers and confirm that, in an organic-matter-, $CaCO_3$-rich sediment
such as in the Equatorial Pacific, the influence of a DBL on diffusive fluxes across the sediment-water interface should indeed
be limited. Additionally, these results confirm the observation by Berelson et al. (1994) that, under $CaCO_3$ and POC deposition
fluxes encountered in the equatorial Pacific, changing the stirring rate in a benthic chamber does not alter steady-state diffusive
fluxes by much. Part of the reason may be the quick adjustment of porewater concentrations to the new diffusive boundary
layer.

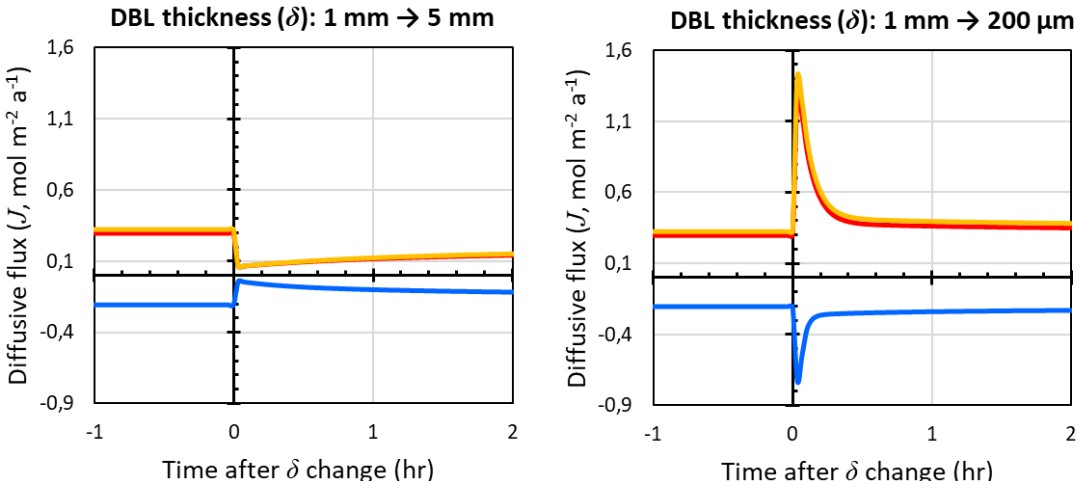

**Figure 9. Response of TAlk, $\Sigma CO_2$, and $O_2$ diffusive fluxes through the DBL to instantaneous changes in the DBL thickness ($\delta$), for instance, as caused by a stirred benthic chamber. Positive values represent solute fluxes toward the bottom waters while negative**
**values represent solute fluxes toward the sediments.**



## 4.4 Additional applications

The time-dependent problems presented above focus on relatively short timescales (from minutes to months). A non-steady-state model such as RADI can also be used to project the sediment response to perturbations over longer periods of
time. Examples include estimating the effect of negative emission technologies such as coastal enhanced weathering with olivine on early diagenesis (Meysman and Montserrat, 2017; Montserrat et al., 2017), of deep-sea mining (Haffert et al., 2020) or bottom trawling (Sala et al., 2021), the impacts of a decadal bottom-water deoxygenation event such as in the Saint Lawrence estuary (Jutras et al., 2020), or the response of sediments to a global-scale ocean acidification event such as the Paleocene–Eocene Thermal Maximum (Zachos et al., 2005; Cui et al., 2011) or the present anthropogenic $CO_2$ transient.

## 775 5 Future developments

One advantage of RADI is that it is easily tunable by the user: adding new components is straightforward as long as the chemical reactions are known. In future releases, we plan to add oxygen, carbon, and calcium isotopes as individual components in order to predict the diagenetic response of isotopic signals. Additionally, adsorption/desorption reactions on clay surfaces could be a critically important advance, especially regarding the prediction of sedimentary pH profiles (Meysman
et al., 2003), as RADI currently treats clay minerals as non-reactive.

The representation of organic matter in the current version of the model is oversimplified. All reactive organic matter in RADI is associated with a 'Redfield' stoichiometry but marine organic matter can considerably deviate from this ideal (Martiny et al., 2013; Teng et al., 2014). In addition, the somewhat arbitrary separation into refractory, slow-decay, and fast-decay pools could be improved by implementing a reactive continuum approach (Middelburg, 1989; Boudreau and Ruddick,
1991), which may be better suited to modelling organic matter degradation over long spatial and temporal scales (Arndt et al., 2013).

Finally, a model is only as good as its assumptions. RADI is targeted to study deep-sea, carbonate sediments. To be used in coastal environments, additional biogeochemical reactions would be necessary, particularly those involving methane and iron sulfide. Close to the shore, sediments become more permeable and the assumption of molecular diffusion as the
dominant mode of solute transport in porewaters does not hold. In very shallow environments that are subject to high wave energy, pressure-induced advection in the sediment porewaters also needs to be included (Huettel et al., 2014).

## Code availability

The current versions of RADI in both Julia and MATLAB/GNU Octave are freely available from GitHub (https://github.com/RADI-model) under the GNU General Public License v3, with more information available online at
https://radi-model.github.io. The exact version of the model used to produce the results used in this paper is archived on Zenodo (RADI.jl v0.3; https://doi.org/10.5281/zenodo.5005650; v1 will be released after review), along with input data and scripts to



run the model for all the simulations presented in this paper. RADI users should cite both this publication and the relevant Zenodo reference (Humphreys and Sulpis, 2021; Sulpis et al., 2021).

**Data availability**

Sediment and porewater composition, porosity, and solid fluxes data for the Southern Pacific Ocean station described in Sayles et al. (2001) are available at http://usjgofs.whoi.edu/jg/dir/jgofs/southern/nbp98_2/. Sediment and porewater composition for the North-western Atlantic Ocean station described in Hales et al. (1994) are available at https://doi.pangaea.de/10.1594/PANGAEA.730420. The GLODAPv2 dataset used in this study is available at https://www.glodap.info/.

**Authors contributions**

**O.S.**: Conceptualization, methodology, software, validation, formal analysis, investigation, writing – original draft, visualization; **M.P.H.**: Conceptualization, methodology, software, formal analysis, investigation, writing – original draft; **M.M.W.**: Conceptualization, methodology, software, formal analysis, writing – review and editing; **D.C.**: Methodology, software, formal analysis, writing – review and editing; **W.M.B.**: Validation, writing – review and editing; **D.M.**: Writing – review and editing, supervision; **J.J.M.**: Methodology, validation, writing – review and editing, supervision; **J.F.A.**: Conceptualization, methodology, validation, resources, writing – review and editing, supervision

**Competing interests**

The authors declare that they have no conflict of interest.

**Acknowledgements**

Thanks are due to Bernard P. Boudreau whose CANDI model (Boudreau, 1996b) was a large source of inspiration during the creation of the present RADI model, and to Daniel L. Johnson for fruitful discussions. We also thank Lukas van de Wiel for assistance with the Utrecht Geoscience computer cluster. O.S. and J.J.M. were supported by the Dutch Ministry of Education via the Netherlands Earth System Science Centre (NESSC). O.S. also acknowledges the Dept. of Earth and Planetary Sciences at McGill University for financial support during his residency in the graduate program and the Faculty of Science at McGill

University for a graduate mobility award. M.M.W, D.C., and D.M., carried out research at the Jet Propulsion Laboratory, California Institute of Technology, under a contract with NASA, with support from the Biological Diversity, Carbon Cycle, Physical Oceanography, and Modeling, Analysis, and Prediction Programs.





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
