# Peer review of "RADIv1: a non-steady-state early diagenetic model for ocean sediments in Julia and MATLAB/GNU Octave"

_Geoscientific Model Development, 2021_

## Referee Comment (RC1)

**Comments on manuscript gmd-2021-211,**
″RADIv1: a non-steady-state early diagenetic model for ocean sediments in Julia and MATLAB/GNU Octave″
by Olivier Sulpis et al.

General comments:

Sulpis et al. presented a new diagenesis model RADI with some examples of its application. The model adopts recently published rate laws for dissolution of calcite and allows for DBL parameterization. The model will be useful to the community especially because it is written in MATLAB and Julia. The manuscript is concisely written and easily understood. However, I have a couple of points of concern regarding the description of model and its performance evaluation and applicability.

1. Model's overall structure

The overall structure of the model is not described in detail. I could find only a few sentences, e.g., L130-132: 'equations composing RADI are based on CANDI, the method-of-lines code by Boudreau (1996b). Unlike the model of Boudreau (1996b), RADI does not solve a set of reactive-transport differential equations but instead computes the concentrations of a set of solids and solutes at each time step following a time vector set by the user.'

Equations provided by the authors, however, indicate that they seem to use some time-forward finite difference method, e.g., L387: 'backward-difference discretization prevails', and thus they seem to have governing differential equations from which difference equations are derived. If so, what is the difference from CANDI with respect to the general model structure including adopted numerical method?

It is somewhat disturbing to read that the model does not solve reactive-transport differential equations (L130-132), and correspondingly the authors did not provide any governing equations. A general reactive-transport equation (e.g., Boudreau, 1996, 1997) formulates the mass conservation law dictating that mass loss/gain via transport and reactions, and mass change within each sediment layer are balanced for each species. Thus, any model (including RADI) should end up in solving reactive-transport equations although the numerical approach can vary with models. Again, however, the numerical method and its difference (if any) from those of other models (including CANDI) are not clearly described in the text.

2. Model evaluation experiments

In steady-state experiments, the authors used Muds to be compared with RADI (Sections 3.1 and 3.2). However, the comparison does not make any sense to me if RADI is tuned but Muds is not, to specific sites considered in this paper.

Also, the diagenetic influence of the updated rate law for $CaCO_3$ dissolution was discussed by comparing Muds and RADI (Section 3.4). This does not make any sense to me, either, given that $\Sigma CO_2$ and TAlk production profiles from OM degradation can be different between the two models. If the authors want to discuss the effect of adopting the new rate law for calcite, they should compare two RADI simulations adopting the new and previous rate laws with individually tuned rate constants under otherwise the same boundary conditions.

3. Model's applicability

The authors argued that RADI can be used for simulations imposing an intense ocean acidification event such as PETM (Section 4.4), but one may doubt it. Under an intense dissolution event (such as PETM), chemical erosion can happen where burial velocity can become negative at certain sediment depths. However, the burial velocity calculation scheme of RADI does not seem to allow this (Eq. (14), Section 2.3 and Fig. 1). Indeed, given that burial calculation does not seem to reflect any mass/volume changes in solid species caused by reactions, the transient simulation of RADI should be limited to the cases where the effect of solid mass/volume changes on burial rate is minor, e.g., short term experiments with minor changes in solid phase concentrations such as those in Sections 4.1-4.3. To enable the application to cases involving a significant $CaCO_3$ dissolution, RADI has to adopt a different burial velocity calculation scheme, such as that adopted by Munhoven (2021, GMD, 14, 3603).

Specific comments:

Fig. 1. Sediment layer/point numbers within the model domain and depths assigned to sediment layers/points seem to be confused. For example, in Fig. 1 Z looks like the total number of sediment points/layers meanwhile it is defined as the total sediment thickness in Table 1 and text. The same goes to dz.

L131-132. Quite vague description of the model. It is unclear how the model is different from or similar to CANDI (please also see general comment 1).

L141. Is the threshold for dz/dt for stability of the numerical solution dependent on w?

L354. Is 30% $d^{-1}$ the most used rate constant with the reaction order of 4.5? I thought 100% $d^{-1}$ is more often adopted in the literature (e.g., Archer, 1991, 1996; Archer et al., 1996).

Eqs. (11)-(14). What is the difference between x and w? Also, in Eq. (13) porewater volume fraction should be used instead of solid volume fraction. How do you calculate $w_\infty$ and $u_\infty$? I guess the authors assume $w_\infty = u_\infty = x_\infty$?

Section 3.1. It does not make any sense to compare 2 models if the 2 models are not tuned to the observations in the same way (only RADI is tuned and Muds is not tuned?). If the rate constants

and/or rate laws are different between the two models, I would expect different boundary conditions for 2 different tuned models.

L528-530. Is the assumption to calculate DBL thickness by Sulpis et al. (2018) consistent with RADI's assumption? In other words, isn't the calculation of DBL thickness by Sulpis et al. (2018) affected by including $CaCO_3$ dissolution by OM-derived $CO_2$?

L549. The good reproduction of $NO_3$ profile seems to be achieved at the cost of bad OM profile reproduction, for which Muds does seem to do a better job. I think the same goes to the $O_2$ profile difference between the two models.

Fig. 4. Calcite concentration does not seem to increase at the bottom, which looks weird given the oversaturation of calcite at the bottom and assuming that RADI allows for precipitation.

Section 3.2. The issues raised for Section 3.1 can apply to Section 3.2.

L649. I guess the authors can definitely tell by repeating the same tuning experiments with RADI adopting the rate law used by Muds. (Please also see general comment 2.) Could you attribute some of differences in $\Sigma CO_2$ and TAlk profiles to the difference in OM diagenesis schemes between the models?

Fig. 8. How can it be possible that $O_2$ and $CO_2$ fluctuate while OM and $CaCO_3$ do not? Is the magnitude of fluctuation of OM and $CaCO_3$ too small to see in the figure?

Section 4.3. Why changes in OM degradation and $CaCO_3$ dissolution are significant while no changes are recognized in Section 4.2? Is this only because imposed DBL changes are larger in Section 4.3? Or is the response of sedimentary system to DBL change dependent on the time rate of DBL change imposed?

Technical corrections:

L71. 'Cappellen' should be 'Van Cappellen'.

Table 1 and throughout. Although the authors stated that variables are written in italic and model notations are in monospaced font (L93), this rule is not completely followed.

---

## Author Comment (AC1)

We thank the reviewer for their time and feedback. Below we reply to comments that asked for information or clarification, and all other suggestions and corrections will be integrated in the revised text as recommended. For clarity, the reviewer's comments are in black, and our replies are in red.

General comments:

Sulpis et al. presented a new diagenesis model RADI with some examples of its application. The model adopts recently published rate laws for dissolution of calcite and allows for DBL parameterization. The model will be useful to the community especially because it is written in MATLAB and Julia. The manuscript is concisely written and easily understood. However, I have a couple of points of concern regarding the description of model and its performance evaluation and applicability.

1. Model's overall structure

The overall structure of the model is not described in detail. I could find only a few sentences, e.g., L130-132: 'equations composing RADI are based on CANDI, the method-of-lines code by Boudreau (1996b). Unlike the model of Boudreau (1996b), RADI does not solve a set of reactive-transport differential equations but instead computes the concentrations of a set of solids and solutes at each time step following a time vector set by the user.' Equations provided by the authors, however, indicate that they seem to use some time-forward finite difference method, e.g., L387: 'backward-difference discretization prevails', and thus they seem to have governing differential equations from which difference equations are derived. If so, what is the difference from CANDI with respect to the general model structure including adopted numerical method?

It is somewhat disturbing to read that the model does not solve reactive-transport differential equations (L130-132), and correspondingly the authors did not provide any governing equations. A general reactive-transport equation (e.g., Boudreau, 1996, 1997) formulates the mass conservation law dictating that mass loss/gain via transport and reactions, and mass change within each sediment layer are balanced for each species. Thus, any model (including RADI) should end up in solving reactive-transport equations although the numerical approach can vary with models. Again, however, the numerical method and its difference (if any) from those of other models (including CANDI) are not clearly described in the text.

The wording describing the model structure was indeed confusing, and we will clarify in the revised manuscript. We will replace the sentence 'Unlike the model of Boudreau (1996b), RADI does not solve a set of reactive-transport differential equations but instead computes the concentrations of a set of solids and solutes at each time step following a time vector set by the user.' by the following:

> "RADI uses the same set of reactive-transport partial differential equations implemented in CANDI (Boudreau, 1996b), i.e., for each solute component,
>
> $$\frac{\partial v}{\partial t} = \frac{1}{\varphi} \frac{\partial}{\partial z}\left(\varphi d \frac{\partial v}{\partial z} - \varphi u v\right) + \alpha(v_w - v) + \sum R$$
>
> and for each solid component,
>
> $$\frac{\partial v}{\partial t} = \frac{1}{\varphi_s} \frac{\partial}{\partial z}\left(\varphi_s b \frac{\partial v}{\partial z} - \varphi_s w v\right) + \sum R$$
>
> where $v$ is the concentration of a given component, $t$ is time, $\varphi$ is porewater porosity and $\varphi_s$ is the solid volume fraction, $d$ is the effective molecular diffusion coefficient and $b$ is the

bioturbation coefficient, *z* is depth, *u* is the porewater burial velocity and *w* is the solid burial velocity, *α* is the irrigation coefficient, $v_w$ is the concentration of a solute in the bottom waters and Σ*R* is the net production rate from all biogeochemical reactions for a given component. Each of these terms will be described in detail later in the Model description section.

These partial differential equations are solved numerically using the method of lines described in Boudreau (1996b). Instead of searching for steady-state solutions directly, RADI computes the concentrations of a set of solids and solutes at each depth and time steps following a time vector set by the user. The user determines the simulation time depending on the objectives, e.g., multimillennial to predict a steady state, or a few days to study the response of the sedimentary system to high frequency cyclic phenomena such as tides."

2. Model evaluation experiments

In steady-state experiments, the authors used Muds to be compared with RADI (Sections 3.1 and 3.2). However, the comparison does not make any sense to me if RADI is tuned but Muds is not, to specific sites considered in this paper.

Also, the diagenetic influence of the updated rate law for CaCO3 dissolution was discussed by comparing Muds and RADI (Section 3.4). This does not make any sense to me, either, given that ΣCO2 and TAlk production profiles from OM degradation can be different between the two models. If the authors want to discuss the effect of adopting the new rate law for calcite, they should compare two RADI simulations adopting the new and previous rate laws with individually tuned rate constants under otherwise the same boundary conditions.

We agree and will address this comment in the revised manuscript, where we will instead compare two RADI simulations for each site, one with the new $CaCO_3$ dissolution kinetics, and one with the high-order $CaCO_3$ dissolution kinetics traditionally used in models, including in Muds.

3. Model's applicability

The authors argued that RADI can be used for simulations imposing an intense ocean acidification event such as PETM (Section 4.4), but one may doubt it. Under an intense dissolution event (such as PETM), chemical erosion can happen where burial velocity can become negative at certain sediment depths. However, the burial velocity calculation scheme of RADI does not seem to allow this (Eq. (14), Section 2.3 and Fig. 1). Indeed, given that burial calculation does not seem to reflect any mass/volume changes in solid species caused by reactions, the transient simulation of RADI should be limited to the cases where the effect of solid mass/volume changes on burial rate is minor, e.g., short term experiments with minor changes in solid phase concentrations such as those in Sections 4.1-4.3. To enable the application to cases involving a significant CaCO3 dissolution, RADI has to adopt a different burial velocity calculation scheme, such as that adopted by Munhoven (2021, GMD, 14, 3603).

The reviewer is correct saying that burial velocity calculations do not reflect changes caused by reactions. In the revised manuscript, we will rewrite and state that RADI cannot deal with long-term major dissolution (erosion) events.

Specific comments:

Fig. 1. Sediment layer/point numbers within the model domain and depths assigned to sediment layers/points seem to be confused. For example, in Fig. 1 Z looks like the total number of sediment

points/layers meanwhile it is defined as the total sediment thickness in Table 1 and text. The same goes to dz.

That is true, we will update Fig. 1 to match the definitions given in the text. *Z* is the total height of the sediment column, and dz is the depth step separating each layer, both are expressed in meters.

L131-132. Quite vague description of the model. It is unclear how the model is different from or similar to CANDI (please also see general comment 1).

We agree and will update the text in the revised manuscript, see reply to comment 1.

L141. Is the threshold for dz/dt for stability of the numerical solution dependent on w?

We have tested this by changing the flux of clay (multiplying and dividing by a hundred) which in returns changes *w*, and it appears that solid burial velocity does not matter for the determination of the dz/dt stability threshold.

L354. Is 30% d-1 the most used rate constant with the reaction order of 4.5? I thought 100% d-1 is more often adopted in the literature (e.g., Archer, 1991, 1996; Archer et al., 1996).

This is correct, a rate constant of 100 % d$^{-1}$ has been used more often in the literature. We will replace in the text and in Fig. 2.

Eqs. (11)-(14). What is the difference between x and w? Also, in Eq. (13) porewater volume fraction should be used instead of solid volume fraction. How do you calculate w∞ and u∞? I guess the authors assume w∞ = u∞ = x∞?

The equations in section contained some inaccuracies as pointed out by the Reviewer. This will be addressed in the revised manuscript. We will rewrite the text and Eqs.(11-14) as:

"The solid burial velocity at the sediment-water interface, $w_0$ in m a$^{-1}$, is given by:

$$w_0 = \frac{\sum \frac{F_v \cdot M_v}{\rho_v}}{\varphi_{s,0}}$$
(...),

where $F_v$ is the flux of a solid species at the sediment-water interface (mol m$^{-2}$ a$^{-1}$), $M_v$ is the molar mass of that solid (g mol$^{-1}$), and $\rho_v$ is its solid density (g m$^{-3}$). The solid and porewater burial velocity at greater depth are assumed to be equal, and are computed as:

$$w_\infty = u_\infty = w_0\, \varphi_{s,0}/\varphi_{s,\infty}$$
(...).

Thus, the porewater burial velocity, *u*, at all depths is:

$$u_z = u_\infty\, \varphi_\infty/\varphi_z$$
(...)

and the solid burial velocity, *w*:

$$w_z = w_\infty\, \varphi_{s,\infty}/\varphi_{s,z}$$
(...)."

This will not affect the model code.

Section 3.1. It does not make any sense to compare 2 models if the 2 models are not tuned to the observations in the same way (only RADI is tuned and Muds is not tuned?). If the rate constants and/or rate laws are different between the two models, I would expect different boundary conditions for 2 different tuned models.

We agree, and will instead compare two RADI simulations for each site, one with the new $CaCO_3$ dissolution kinetics, and one with the high-order $CaCO_3$ dissolution kinetics traditionally used in models, including in Muds. We hope this will address this comment.

L528-530. Is the assumption to calculate DBL thickness by Sulpis et al. (2018) consistent with RADI's assumption? In other words, isn't the calculation of DBL thickness by Sulpis et al. (2018) affected by including CaCO3 dissolution by OM-derived CO2?

In Sulpis et al. (2018) the DBL thickness is computed as a function of the bottom-water current speed and of the viscosity of seawater, and does not depend on any assumption made regarding organic matter degradation or CaCO3 dissolution. Hence this method to calculate DBL thickness is consistent with RADI's assumptions.

L549. The good reproduction of NO3 profile seems to be achieved at the cost of bad OM profile reproduction, for which Muds does seem to do a better job. I think the same goes to the O2 profile difference between the two models.

That is correct, POC concentrations are always better predicted by Muds than by RADI near the sediment surface, although it seems to be the opposite at depths.

Fig. 4. Calcite concentration does not seem to increase at the bottom, which looks weird given the oversaturation of calcite at the bottom and assuming that RADI allows for precipitation.

Calcite concentration does increase at depth but very slightly, by about 0.1 wt%. That is because, at 40 cm below the sediment surface, the predicted saturation state of porewater with respect to calcite is only at a value of ~1.05. We will precise that in the revised manuscript.

Section 3.2. The issues raised for Section 3.1 can apply to Section 3.2. L649. I guess the authors can definitely tell by repeating the same tuning experiments with RADI adopting the rate law used by Muds. (Please also see general comment 2.) Could you attribute some of differences in ⬚CO2 and TAlk profiles to the difference in OM diagenesis schemes between the models?

We agree, and will instead compare two RADI simulations for each site, one with the new $CaCO_3$ dissolution kinetics, and one with the high-order $CaCO_3$ dissolution kinetics traditionally used in models, rather than comparing with Muds directly without being able to attribute the differences to a process in particular.

Fig. 8. How can it be possible that O2 and CO2 fluctuate while OM and CaCO3 do not? Is the magnitude of fluctuation of OM and CaCO3 too small to see in the figure?

That is correct, [POC] and [Calcite] fluctuate too but the magnitude of the fluctuation is infinitely small. When, every 6 hours, the DBL is thinning for 3 hours to do the increasing tidal current speed, solutes can diffuse in or out of the sediment quicker, but solids do not have time to adjust due to the very slow accumulation rates in the deep sea.

Section 4.3. Why changes in OM degradation and CaCO3 dissolution are significant while no changes are recognized in Section 4.2? Is this only because imposed DBL changes are larger in Section 4.3? Or is the response of sedimentary system to DBL change dependent on the time rate of DBL change imposed?

On the timescale of this benthic chamber simulation (2 hours), changes in the POC degradation and $CaCO_3$ dissolution rates are actually also negligible, what is shown in Fig. 9 are TAlk (in red), $\Sigma CO_2$ (in yellow) and $O_2$ (in blue) fluxes. This was not labelled properly and we will add this to the figure in the

revised version. We do not impose the response of the sediment system to DBL changes, the porewater composition adjusts to DBL changes. Please note that we are currently working on a follow-up RADI study focusing on the competitive control of organic matter degradation and the DBL on $CaCO_3$ dissolution at the seafloor, in which we will discuss this in details and provide in-depth analyses.

We accept all the other suggestions and corrections, and will include them in the revised manuscript.

Brest regards,

Olivier Sulpis, Matthew P. Humphreys, Monica M. Wilhelmus, Dustin Carroll, William M. Berelson, Dimitris Menemenlis, Jack J. Middelburg, Jess F. Adkins

**References**

Boudreau, B.P. A method-of-lines code for carbon and nutrient diagenesis in aquatic sediments. Computers & Geosciences, 22, 479-496, 1996b

Sulpis, O., Boudreau, B.P., Mucci, A., Jenkins, C.J., Trossman, D.S., Arbic, B.K. and Key, R.M.: Current $CaCO_3$ dissolution at the seafloor caused by anthropogenic $CO_2$. Proceedings of the National Academy of Sciences, 115, 11700-11705, 2018

---

## Author Comment (AC2)

We thank David Burdige for his time and feedback. Below we reply to comments that asked for information or clarification, and all other suggestions and corrections will be integrated in the revised text as recommended. For clarity, David Burdige's comments are in black, and our replies are in red.

In this manuscript Sulpis et al. describe a diagenetic model for marine sediments targeted for the study of deep-sea sediments (RADI). Many aspects of this model look like a number of other models that have been previously published, although there are two aspects of this work that stand out. The first is that the model incorporates a more complex (and presumably realistic) representation of carbonate dissolution kinetics, based on recent studies from the USC and CalTech groups. This is an important addition for a number of reasons and warrants eventual publication of this work.

The second is that the model is explicitly discussed in the context of its ability to look at time-dependent problems. Strictly speaking, this is not exactly new since other models out there (e.g., CANDI) can, in principle, also be used to examine time-dependent problems. However, these models generally are not used in this fashion and the work shown here presents some interesting observations based on time-dependent simulations using RADI.

A concern I have, though, about the model is that they use total alkalinity and total DIC as solute variables, rather than calculating them from individual chemical components (I know this is touted as an advantage of this model, but I'm not as convinced that it really is). Given that a major thrust of this work is examining carbonate dissolution in deep-sea sediments this seems like a possible problem. Rather than calculating [H+] and [CO32-] from Alk and DIC profiles (see line 297) wouldn't it make more sense to model H+ and carbonate (or bicarbonate) concentrations directly in the model and then calculate alkalinity and DIC depth profiles at the end of each time step. Such a model would be just as easy to use as RADI is in terms of comparing model results with field observations and would likely be more accurate. This would probably also require the addition of an equation for borate in the model, but that would be an easy addition (see, for example, the approaches described in Hoffmann et al. [Biogeosci. 5, 227-251, 2008] and Faber et al. [Biogeosci. 9, 4087-4097, 2012]).

Transport-reaction modelling of the carbonate system is challenging, and trade-offs must be made (e.g., Boudreau, 1997; Hoffmann et al., 2008 Biogeosciences). Modelling the carbonate system in the water column normally involves transport of total alkalinity and total DIC because these model variables are conservative (Humphreys et al., 2018 Marine Chemistry), and both are transported with the water (i.e. differences in molecular diffusion among the species involved do not matter in advection dominated systems). In non-permeable sediments, molecular diffusion dominates transport and species diffuse according to their own gradient (e.g. the gradient of carbon dioxide may differ from that of the carbonate ion). Moreover, species may have different diffusion coefficients. As a consequence, a more accurate way to represent the carbonate system in non-permeable sediments is to have transport-reaction equations for all the species involved (protons, bicarbonate, carbonate, carbon dioxide) as suggested by David Burdige. The most accurate way would be to include also changes in complex formation and electroneutrality constraints (Boudreau et al., 2004 EPSL).

However, this gain in model accuracy comes at the expense of model complexity and computational demand, while the uncertainty due to lumping carbonic acid, bicarbonate and carbonate transport is usually less than other uncertainties related to thermodynamic constants needed in the calculation and tortuosity corrections for diffusion coefficients. Most of the uncertainty surrounding $CaCO_3$ saturation state estimates stems from the large uncertainty within the $CaCO_3$ equilibrium constants

rather than that associated with carbonate ion (Orr et al., 2018 Marine Chemistry; Sulpis et al., 2020 Ocean Science); this uncertainty is independent of the modelling approach used.

$HCO_3^-$ is the dominant dissolved inorganic carbon species at pH values typical of porewaters, see figure below on the left. In the figure below, on the right, we show what the diffusion of DIC would be if it was computed as a weighted arithmetic average of the diffusion coefficients of each of the DIC components. At pH ~7, the 'true' DIC diffusion coefficient (the weighted arithmetic average) is about 4% higher than the diffusion coefficient of $HCO_3^-$ alone. That gap becomes negligible at higher pH. Thus, using the $HCO_3^-$ diffusion coefficient as a surrogate for that of TAlk and DIC, as it has been done by other, similar models, such as that of Kanzaki et al. (2021, Geoscientific Model Development), should be acceptable as long as the pH is not too low. We will include this figure and discussion in the supplementary information.

[Figure]

Borate is already included in the model for solving the $CO_2$ system. We have not made that clear in the current version of the manuscript, but will include the following in the next version, line 298.

"At each model time step, the total hydrogen ion concentration $[H^+]$ is computed from TAlk and $\sum CO_2$ using a single Newton-Raphson iteration from the previous time step (Humphreys et al., 2021, Geosci Model Dev Discuss):

$$[H^+]_t = [H^+]_{t-1} - \frac{[TAlk]([H^+]_{t-1}, [\sum CO_2]) - [TAlk]}{d[TAlk]([H^+]_{t-1}, [\sum CO_2]) \Big/ d[H^+]_{t-1}}$$

where $[H^+]_t$ is the new $[H^+]$ value and $[H^+]_{t-1}$ is the $[H^+]$ from the previous time step. $TAlk([H^+]_{t-1}, \sum CO_2)$ is the total alkalinity computed from user-specified total dissolved silicate, $[\sum PO_4]$ and total borate calculated from salinity (Uppstrom 1974), plus equilibrium constants for silicic acid (Sillen et al. 1964, Special publication of the Royal Society of Chemistry) and phosphoric acid (Yao and Millero 1995, Aquatic Geochemistry). Its derivative is computed following the approach of CO2SYS, see Humphreys et al. (2021, Geoscientific Model Development Discussions). The carbonate ion concentration is then computed as:

$$[CO_3^{2-}] = \frac{[\sum CO_2] \times K_1^* \times K_2^*}{K_1^* \times K_2^* + K_1^* \times [H^+]_t + [H^+]_t^2}$$

> where $K_1^*$ and $K_2^*$ are the first and second dissociation constants for carbonic acid, respectively, taken from Lueker et al. (2000, Marine Chemistry)."

Maybe I'm missing something, but since one of the major efforts here is to more realistically quantify carbonate dissolution in early diagenetic models the approach taken in RADI to "directly" model Alkalinity and DIC as solute variables seems like it adds unnecessary uncertainty. It also requires that they use the bicarbonate diffusion coefficient for DIC and alkalinity (line 395), which adds further uncertainty in the calculation of carbonate ion concentration gradients and fluxes near the sediment surface, which is where most calcium carbonate dissolves in the deep sea.

See previous reply.

I also have a number of other questions and smaller concerns about this work. Note that in the rest of this review, the numbers in parentheses refer to line numbers.

(130-1) – Just so I'm clear, the "diagenetic equations" in RADI are still differential equations (also see the next comment).

That is correct, the reviewer #1 made a similar comment. We have clarified the text, and added partial differential equations ruling the model, see reply to comment #1 of reviewer #1.

(132) – How exactly does RADI compute "the concentrations of a set of solids and solutes at each time step"? This should be briefly discussed here and perhaps presented in a bit more detail in the supporting information section. Also, in addition to describing these equations part-by-part in the text it might be nice to present some sort of summary of the complete equations in the supporting information section. Perhaps only modeling "geeks" like me would want to see this, but I think it would be good for this to be available, should any reader be so interested in seeing this information.

We have clarified and complemented this section and included the model governing equation, in response to the first comment of reviewer #1. Eq. (3) is used to compute the new concentrations at each time step, and each of the R, A, D & I terms in Eq. (3) are defined later in the Model description section. In addition, we will include a summary in the supplement that will include the complete, detailed version of Eq. (3).

(181) – Why and how is Clay being modeled here? This is never really discussed (unless I missed it).

Clay is simply modelled as a non-reactive solid. We use it to assess when the simulation has reached a steady state, and because the clay accumulation flux to the sediment-water interface participates to the calculation of the solid burial velocity. We will precise this at line 181. In future versions, we plan to incorporate adsorption/desorption reactions on clay surfaces, as mentioned in section 5.

(228) - What exactly is meant by "the reactivities decline with depth"? I'm assuming the k's in eqn. (4) are constants for a given site (since based on eqns. (7a) and (7b) they only depend on the carbon rain rate to the sediment), so the overall rate of organic matter degradation at any given site should decrease with depth solely because of changes with depth in the quantity of organic matter and the relative proportions of fast and slow decay materials that are present. Am I missing something here?

This is a mistake, reactivities are actually constant with depth, as computed per equations (7a,7b). We will delete this part of the sentence.

(250) – Is there a subscript "z" missing for the k on this line?

Here and elsewhere, because the organic matter reactivities do not decrease with depth, there should not be any subscript 'z'. We will delete this.

(268) – "This scheme …" - I would think that the way alkalinity and DIC are modeled in RADI will also make it difficult to apply this model to coastal and anoxic sediments.

We agree that below at lower pH, using the $HCO_3^-$ diffusion coefficient for DIC and TAlk may not be adequate. We will precise that and refer to the new supplementary figure showed above in response to comment #2.

(299) – How are the concentrations of fluoride, borate and silicate obtained?

As mentioned in the reply to comment #1, we have added further information to *CaCO_3 dissolution and precipitation* section 2.2.3. Borate concentration was computed as a function of salinity (Uppstrom, 1974, DSR) for borate. Fluorite is not included in the current version of RADI. The silicate concentration was fixed to a typical deep-sea value of 120 µmol $kg^{-1}$.

(307) – It's not clear to me how the reaction order (η) implicitly accounts for each carbonate mineral's specific surface area. The discussion later on only talks about how the reaction order varies with Ω.

The sentence was unclear, in fact, only the dissolution rate constant accounts for the mineral surface area. The reaction order is specific to each mineral and only depends on Ω. We will rephrase to clarify.

In the caption to Fig. 2 it states that calcite and aragonite dissolution rates are based on eqns. (8) and (9) but it isn't until you read a little further that you learn that η is actually a complex function of Ω (starting on line 334). It might be good to at least mention this in the figure caption.

We will clarify this in the figure caption.

Fig. 4 has two different types of symbols in the O2 profiles. I think I figured out the differences from reading the text (lines 547-548), but this should be clarified in the figure legend and/or caption.

The squares refer to [$O_2$] from microelectrodes and the circles refer to [$O_2$] from porewater extracted from the retrieved box core. We will precise that in the caption.

(583) – "in the laboratory" but on-board ship? Please clarify.

Yes, that is indeed from the ship laboratory. We will clarify.

(704) - Will tides in the deep sea (water depth 4370 at station #W-2) really generate this large a change in DBL thickness (i.e., from 0.5 to 3.5 mm)?

There was a mistake in the text, in the simulation, the DBL actually varied between 0.5 and 1.5 mm-thick, as shown in Fig. 8. We will correct this in the revised manuscript. There is unfortunately no data to assess the validity of this deep-sea tidal DBL-range. Assuming that the DBL thickness depends on the current speed only, and that the seafloor is perfectly flat, i.e., no turbulence generated by sediment roughness, without tidal current, a DBL value of 1.5 mm-thick could be realistic in abyssal plains (Sulpis et al., 2018, PNAS). Since it is thought that tidal currents account for most of the overall current speed in the deep-sea, that the DBL thins to 0.5 mm every 6 hours when the tidal current is the strongest could be realistic.

(715) – Tides affect both the concentration gradient at the sediment water interface as well as the thickness of the DBL, so the combined effects should then impact the benthic flux. Do these two factors reinforce one another, cancel each other out, or do something in-between? Some of these issues are alluded to indirectly in the next section (4.3), but I wonder if this is worth looking at and/or considering here more explicitly.

This is correct. Regarding the diffusive fluxes across the sediment-water interface, interface concentration changes and DBL thickness change have generally opposite effects. Interfacial concentrations respond quickly to DBL changes, but that the gradient within the DBL also changes will attenuate the initial response. This is what is seen in section 4.3., and we will include a few pH depth profiles (see reply to next comment) to illustrate this effect. We will not discuss that in detail here because that will be the object of a manuscript currently in preparation, in which we will investigate the competing effects of DBL changes and organic matter degradation on $CaCO_3$ dissolution near the sediment-water interface.

(760-1) – "… the quick adjustment of porewater concentrations to the new diffusive boundary layer …" – Might it be worth showing these profiles (perhaps in the supporting information section)?

We will include these pH profiles in the revised supplement.

(781) – I think that trying to model carbon isotopes from POC degradation will be challenging if you use a reactive continuum approach, even if this approach is more appropriate for many studies. In part, that is why we use a similar "multi-G" approach to the one used here in our work modeling POC, DIC and DOC degradation in sediments (both total C, 13C and 14C)

We will delete the statement about switching to a continuum approach.

(787) – As noted above, I wonder if applications of the RADI model to coastal sediments will be difficult given the way alkalinity and DIC are being modelled here.

We will add a note precising that if pH is too low, assuming that both DIC and TAlk can diffuse at the same rate than a $HCO_3^-$ will lead to inaccurate results.

We accept all the other suggestions and corrections and will include them in the revised manuscript.

Best regards,

Olivier Sulpis, Matthew P. Humphreys, Monica M. Wilhelmus, Dustin Carroll, William M. Berelson, Dimitris Menemenlis, Jack J. Middelburg, Jess F. Adkins

**References**

Boudreau, B.P.: Diagenetic Models and Their Implementation. Springer-Verlag, Berlin, 414 pp. 1997

Boudreau, B.P., Meysman, F.J.R., Middelburg, J.J. Multicomponent ionic diffusion in porewaters: Coulombic effects revisited. Earth and Planetary Science Letters, 222, 653-666, 2004

Hofmann, A.F., Meysman, F.J.R., Soetaert, K., Middelburg, J.J. A step-by-step procedure for pH model construction in aquatic systems. Biogeosciences 5, 227-251, 2008

Humphreys, M.P., Daniels, C.J., Wolf-Gladrow, D.A., Tyrrell, T., Achterberg, E.P. On the influence of marine biogeochemical processes over $CO_2$ exchange between the atmosphere and ocean. Marine Chemistry, 199, 1-11, 2018

Humphreys, M.P., Lewis, E.R., Sharo, J.D., Pierrot, D. PyCO2SYS v1.7: marine carbonate system calculations in Python. Geoscientific Model Development Discussions. 2021

Lueker, T. J., Dickson, A. G., and Keeling, C. D.: Ocean $pCO_2$ calculated from dissolved inorganic carbon, alkalinity, and equations for K1 and K2: validation based on laboratory measurements of $CO_2$ in gas and seawater at equilibrium, Mar. Chem., 70, 105–119, 2000

Kanzaki, Y., Hülse, D., Turner, S.K., Ridgwell, A. A model for marine sedimentary carbonate diagenesis and paleoclimate proxy signal tracking: IMP v1.0. Geoscientific Model Development, 14, 5999-6023, 2021

Orr, J. C., Epitalon, J.-M., Dickson, A. G., and Gattuso, J.-P.: Routine uncertainty propagation for the marine carbon dioxide system, Mar. Chem., 207, 84–107, 2018

Sillén, L.G., Martell, A.E., Bjerrum, J., Schwarzenbach, G.K. Stability Constants of metal-ion complexes. The Chemical Society (London), Special Publ. 17:751, 1964

Sulpis, O., Boudreau, B.P., Mucci, A., Jenkins, C.J., Trossman, D.S., Arbic, B.K. and Key, R.M.: Current $CaCO_3$ dissolution at the seafloor caused by anthropogenic $CO_2$. Proceedings of the National Academy of Sciences, 115, 11700-11705, 2018

Sulpis, O., Lauvset, S.K., Hagens, M. Current estimates of $K_1^*$ and $K_2^*$ appear inconsistent with measured $CO_2$ system parameters in cold oceanic regions. Ocean Science 16(4), 847-862, 2020

Uppström, L. R.: The boron/chlorinity ratio of deep-sea water from the Pacific Ocean, Deep-Sea Research and Oceanographic Abstracts, 21, 161-162, 1974

Yao, W. and Millero, F. J.: The chemistry of the anoxic waters in the Framvaren Fjord, Norway, Aquat. Geochem., 1, 53–88, 1995

---

## Author Response (AR1)

Dear Dr. Yool,

Please find below our listing of the changes in response to the reviewers' comments. We refer to our author comments at https://doi.org/10.5194/gmd-2021-211-AC1 and https://doi.org/10.5194/gmd-2021-211-AC2 for our detailed responses. A tracked-change version of the manuscript highlighting the insertions and deletions in the text has also been uploaded alongside the revised manuscript. We hope these changes will address the reviewer's comments.

Best regards,

Olivier Sulpis, on behalf of the authors
* * *
**Revisions in response to comments by Anonymous Referee #1**

The overall structure of the model is not described in detail. I could find only a few sentences, e.g., L130-132: 'equations composing RADI are based on CANDI, the method-of-lines code by Boudreau (1996b). Unlike the model of Boudreau (1996b), RADI does not solve a set of reactive-transport differential equations but instead computes the concentrations of a set of solids and solutes at each time step following a time vector set by the user.' Equations provided by the authors, however, indicate that they seem to use some time-forward finite difference method, e.g., L387: 'backward difference discretization prevails', and thus they seem to have governing differential equations from which difference equations are derived. If so, what is the difference from CANDI with respect to the general model structure including adopted numerical method?

It is somewhat disturbing to read that the model does not solve reactive-transport differential equations (L130-132), and correspondingly the authors did not provide any governing equations. A general reactive-transport equation (e.g., Boudreau, 1996, 1997) formulates the mass conservation law dictating that mass loss/gain via transport and reactions, and mass change within each sediment layer are balanced for each species. Thus, any model (including RADI) should end up in solving reactive-transport equations although the numerical approach can vary with models. Again, however, the numerical method and its difference (if any) from those of other models (including CANDI) are not clearly described in the text.

- **We added some text to section 2.1. and added two new equations describing the reactive-transport partial differential equations.**

In steady-state experiments, the authors used Muds to be compared with RADI (Sections 3.1 and 3.2). However, the comparison does not make any sense to me if RADI is tuned but Muds is not, to specific sites considered in this paper.

Also, the diagenetic influence of the updated rate law for CaCO3 dissolution was discussed by comparing Muds and RADI (Section 3.4). This does not make any sense to me, either, given that $\Sigma CO2$ and TAlk production profiles from OM degradation can be different between the two models. If the authors want to discuss the effect of adopting the new rate law for calcite, they should compare two RADI simulations adopting the new and previous rate laws with individually tuned rate constants under otherwise the same boundary conditions.

- **We updated the text in section 3, Fig. 4 and Fig. 5. We do not use Muds anymore and instead compare two RADI simulations, one with 'traditional' and one with 'new' calcium carbonate kinetics.**

The authors argued that RADI can be used for simulations imposing an intense ocean acidification event such as PETM (Section 4.4), but one may doubt it. Under an intense dissolution event (such as PETM), chemical erosion can happen where burial velocity can become negative at certain sediment depths. However, the burial velocity calculation scheme of RADI does not seem to allow this (Eq. (14), Section 2.3 and Fig. 1). Indeed, given that burial calculation does not seem to reflect any mass/volume changes in solid species caused by reactions, the transient simulation of RADI should be limited to the cases where the effect of solid mass/volume changes on burial rate is minor, e.g., short term experiments with minor changes in solid phase concentrations such as those in Sections 4.1-4.3. To enable the application to cases involving a significant CaCO3 dissolution, RADI has to adopt a different burial velocity calculation scheme, such as that adopted by Munhoven (2021, GMD, 14, 3603).

- **In section 4.4., we deleted the mention to the PETM and added some text to explain that a different burial velocity calculation scheme would need to be implemented to deal with long term major dissolution events.**

Fig. 1. Sediment layer/point numbers within the model domain and depths assigned to sediment layers/points seem to be confused. For example, in Fig. 1 Z looks like the total number of sediment points/layers meanwhile it is defined as the total sediment thickness in Table 1 and text. The same goes to dz.

- **We updated Fig. 1 to reflect definitions given in the text.**

L131-132. Quite vague description of the model. It is unclear how the model is different from or similar to CANDI (please also see general comment 1).

- **We added some text to section 2.1. and added two new equations describing the reactive-transport partial differential equations.**

L141. Is the threshold for dz/dt for stability of the numerical solution dependent on w?

- **No change required.**

L354. Is 30% d-1 the most used rate constant with the reaction order of 4.5? I thought 100% d-1 is more often adopted in the literature (e.g., Archer, 1991, 1996; Archer et al., 1996).

- **We changed the rate constant to 100% d$^{-1}$ and have updated the text, the figures, and the model simulations presented in section 3 to reflect this new value.**

Eqs. (11)-(14). What is the difference between x and w? Also, in Eq. (13) porewater volume fraction should be used instead of solid volume fraction. How do you calculate w$\infty$ and u$\infty$? I guess the authors assume w$\infty$ = u$\infty$ = x$\infty$?

- **We changed the text and equations in section 2.3 to correct for the mistakes pointed out to by the reviewer.**

Section 3.1. It does not make any sense to compare 2 models if the 2 models are not tuned to the observations in the same way (only RADI is tuned and Muds is not tuned?). If the rate constants and/or

rate laws are different between the two models, I would expect different boundary conditions for 2 different tuned models.

- **We updated the text in section 3, Fig. 4 and Fig. 5. We do not use Muds anymore and instead compare two RADI simulations, one with 'traditional' and one with 'new' calcium carbonate kinetics.**

L528-530. Is the assumption to calculate DBL thickness by Sulpis et al. (2018) consistent with RADI's assumption? In other words, isn't the calculation of DBL thickness by Sulpis et al. (2018) affected by including CaCO3 dissolution by OM-derived CO2?

- **No change required.**

L549. The good reproduction of NO3 profile seems to be achieved at the cost of bad OM profile reproduction, for which Muds does seem to do a better job. I think the same goes to the O2 profile difference between the two models

- **No change required.**

Fig. 4. Calcite concentration does not seem to increase at the bottom, which looks weird given the oversaturation of calcite at the bottom and assuming that RADI allows for precipitation.

- **We added a new sentence at the end of section 3.1 discussing the calcite increase at the bottom.**

Section 3.2. The issues raised for Section 3.1 can apply to Section 3.2. L649. I guess the authors can definitely tell by repeating the same tuning experiments with RADI adopting the rate law used by Muds. (Please also see general comment 2.) Could you attribute some of differences in CO2 and TAlk profiles to the difference in OM diagenesis schemes between the models?

- **We updated the text in section 3, Fig. 4 and Fig. 5. We do not use Muds anymore and instead compare two RADI simulations, one with 'traditional' and one with 'new' calcium carbonate kinetics.**

Fig. 8. How can it be possible that O2 and CO2 fluctuate while OM and CaCO3 do not? Is the magnitude of fluctuation of OM and CaCO3 too small to see in the figure?

- **No change required.**

Section 4.3. Why changes in OM degradation and CaCO3 dissolution are significant while no changes are recognized in Section 4.2? Is this only because imposed DBL changes are larger in Section 4.3? Or is the response of sedimentary system to DBL change dependent on the time rate of DBL change imposed?

- **We edited Fig. 9 which was not labelled properly.**

L71. 'Cappellen' should be 'Van Cappellen'

- **We changed that.**

Table 1 and throughout. Although the authors stated that variables are written in italic and model notations are in monospaced font (L93), this rule is not completely followed.

- **We updated Table 1.**

**Revisions in response to comments by Dr. Burdige**

The second is that the model is explicitly discussed in the context of its ability to look at time dependent problems. Strictly speaking, this is not exactly new since other models out there (e.g., CANDI) can, in principle, also be used to examine time-dependent problems. However, these models generally are not used in this fashion and the work shown here presents some interesting observations based on time-dependent simulations using RADI.

A concern I have, though, about the model is that they use total alkalinity and total DIC as solute variables, rather than calculating them from individual chemical components (I know this is touted as an advantage of this model, but I'm not as convinced that it really is). Given that a major thrust of this work is examining carbonate dissolution in deep-sea sediments this seems like a possible problem. Rather than calculating [H+] and [CO32-] from Alk and DIC profiles (see line 297) wouldn't it make more sense to model H+ and carbonate (or bicarbonate) concentrations directly in the model and then calculate alkalinity and DIC depth profiles at the end of each time step. Such a model would be just as easy to use as RADI is in terms of comparing model results with field observations and would likely be more accurate. This would probably also require the addition of an equation for borate in the model, but that would be an easy addition (see, for example, the approaches described in Hoffmann et al. [Biogeosci. 5, 227-251, 2008] and Faber et al. [Biogeosci. 9, 4087-4097, 2012]).

Maybe I'm missing something, but since one of the major efforts here is to more realistically quantify carbonate dissolution in early diagenetic models the approach taken in RADI to "directly" model Alkalinity and DIC as solute variables seems like it adds unnecessary uncertainty. It also requires that they use the bicarbonate diffusion coefficient for DIC and alkalinity (line 395), which adds further uncertainty in the calculation of carbonate ion concentration gradients and fluxes near the sediment surface, which is where most calcium carbonate dissolves in the deep sea.

- **We added one new discussion section in the supplement ("Using TAlk and ΣCO2 as solute variables, rather than their individual components"), one supplementary figure (Fig. S2), some text and two new equations in section 2.2.3. We have also added a sentence in section 2.4 discussing this and referring to Fig. S2.**

(130-1) – Just so I'm clear, the "diagenetic equations" in RADI are still differential equations (also see the next comment) + (132) – How exactly does RADI compute "the concentrations of a set of solids and solutes at each time step"? This should be briefly discussed here and perhaps presented in a bit more detail in the supporting information section. Also, in addition to describing these equations part-by-part in the text it might be nice to present some sort of summary of the complete equations in the supporting information section. Perhaps only modeling "geeks" like me would want to see this, but I think it would be good for this to be available, should any reader be so interested in seeing this information.

- **We added some text to section 2.1. and added two new equations describing the reactive-transport partial differential equations, we have also added a new supplementary section ("Full RADI equations").**

(181) – Why and how is Clay being modeled here? This is never really discussed (unless I missed it).

- **We added a sentence precising that at the end of section 2.1.**

(228) - What exactly is meant by "the reactivities decline with depth"? I'm assuming the k's in eqn. (4) are constants for a given site (since based on eqns. (7a) and (7b) they only depend on the carbon rain rate

to the sediment), so the overall rate of organic matter degradation at any given site should decrease with depth solely because of changes with depth in the quantity of organic matter and the relative proportions of fast and slow decay materials that are present. Am I missing something here?

- **We edited that part of the text in section 2.2.1 and deleted the mention that reactivities decrease with depth.**

(250) – Is there a subscript "z" missing for the k on this line?

- **We edited equations (6), (9a) and (9b).**

(268) – "This scheme …" - I would think that the way alkalinity and DIC are modeled in RADI will also make it difficult to apply this model to coastal and anoxic sediments

- **We added a sentence at the end of section 5 to make this point.**

(299) – How are the concentrations of fluoride, borate and silicate obtained?

- **We updated the text in section 2.2.3 to remove the mention to fluoride and explain where borate and silicate concentrations are taken from.**

(307) – It's not clear to me how the reaction order ($\eta$) implicitly accounts for each carbonate mineral's specific surface area. The discussion later on only talks about how the reaction order varies with $\Omega$.

- **We rephrased that sentence to clarify.**

In the caption to Fig. 2 it states that calcite and aragonite dissolution rates are based on eqns. (8) and (9) but it isn't until you read a little further that you learn that $\eta$ is actually a complex function of $\Omega$ (starting on line 334). It might be good to at least mention this in the figure caption.

- **We clarified that in the figure caption.**

Fig. 4 has two different types of symbols in the O2 profiles. I think I figured out the differences from reading the text (lines 547-548), but this should be clarified in the figure legend and/or caption

- **We clarified that in the legend.**

(583) – "in the laboratory" but on-board ship? Please clarify.

- **We clarified that in the sentence.**

(704) - Will tides in the deep sea (water depth 4370 at station #W-2) really generate this large a change in DBL thickness (i.e., from 0.5 to 3.5 mm)?

- **We changed those values to reflect those that were used in the simulation and in Fig. 8, that span a narrower, more realistic range.**

(715) – Tides affect both the concentration gradient at the sediment water interface as well as the thickness of the DBL, so the combined effects should then impact the benthic flux. Do these two factors reinforce one another, cancel each other out, or do something in-between? Some of these issues are alluded to indirectly in the next section (4.3), but I wonder if this is worth looking at and/or considering here more explicitly.

- **No change required.**

(753) – I would say "a 5-fold decrease in δ".

- **We changed that.**

(760-1) – "… the quick adjustment of porewater concentrations to the new diffusive boundary layer …" – Might it be worth showing these profiles (perhaps in the supporting information section)?

- **We added these profiles in a new figure in the supplementary information (Fig. S3).**

9 – Please define which color goes with which solute. I'm assuming blue is oxygen, but this (and the other colors) should be explicitly labelled.

- **We updated Fig. 9.**

(781) – I think that trying to model carbon isotopes from POC degradation will be challenging if you use a reactive continuum approach, even if this approach is more appropriate for many studies. In part, that is why we use a similar "multi-G" approach to the one used here in our work modeling POC, DIC and DOC degradation in sediments (both total C, 13C and 14C)

- **We have deleted the sentence mentioning a reactive continuum approach.**

(787) – As noted above, I wonder if applications of the RADI model to coastal sediments will be difficult given the way alkalinity and DIC are being modelled here

- **We added a sentence at the end of section 5 to make this point.**

---

## Author Response (AR2)

**Reviewer #1**

I think the authors mostly addressed my comments from the first review.

The manuscript now shows the effect of the adopted new rate law more clearly (but I have one additional comment on the comparison of different rate laws and parameter tuning; please see general comments below). The other emphasized features of the model (e.g., time-dependency, 3G for OM and diffusive boundary) are novel but not necessarily new and the model's application is still limited, e.g., not ready for ocean acidification simulation. Nonetheless illustrated examples are interesting and the important aspects of early diagenetic modeling are concisely described. Given the detailed description of adopted rate laws and their tuning and because the model is written in Julia and MATLAB, the paper will be useful to the community. Overall, I think this version of manuscript could be acceptable for publication in GMD.

General comments:

Comparison of the two models with the new and traditional rate laws, however, is still arbitrary because the rate constants of both models are generally determined by tuning. One cannot tell whether the difference stated in the manuscript may disappear or not when adopting different rate constants for the two models. If the feature of the new rate law is the saturation-independent enhanced dissolution caused by step-edge retreat close to equilibrium (Section 2.2.3) and the authors want to quantify its effect, then the authors should adopt the same far-from-equilibrium rate constants for both models assuming that both are controlled by the same dissolution mechanisms at far from equilibrium. This helps suppress any artificial difference caused by different tuning by different authors and make the comparison clearer and more meaningful with respect to dissolution mechanisms. This may be done by changing the rate constants of either model, so that curves overlap one another in far-from-equilibrium region in Fig. 2.

We thank the reviewer for this positive review and for providing critical comments. We agree with the reviewer that to make the comparison more meaningful with respect to dissolution mechanisms, the rate constant of the "traditional" calcite dissolution rate law should be tuned so that it overlaps with the far-from-equilibrium dissolution rate law adopted in RADI. We have changed the dissolution rate constant for the "traditional" law from 100 % day$^{-1}$ to 10 % day$^{-1}$ so that far from equilibrium (at $\Omega_{ca}$ ~ 0.6, see Fig. 2), dissolution rates from both laws are equal.

We have updated the model the Model evaluation sections 3.1 and 3.2 which now show results from simulations using a "traditional" rate law with a 10 % day$^{-1}$ rate constant. In both the north Atlantic and the southern Pacific stations, the calcite contents predicted by both rate laws are now similar. To describe this and include these updated results, we have also changed the following:

- We have updated Fig. 2, Fig. 4 and Fig. 5
- [section 2.2.3, last paragraph, added text]: The value of 10 % day-1 for the rate constant was chosen because it makes the "traditional" calcite dissolution law overlap with the RADI dissolution law, so that any differences between the two can be attributed to enhanced dissolution caused by step-edge retreat close to equilibrium.
- [section 3.1, last paragraph, added text]: When "traditional" 4.5-order calcite dissolution kinetics are implemented, calcite concentrations are similar to those predicted by RADI, but the predicted [TAlk] and [∑CO2] are slightly different, being lower (< ~10 µmol kg$^{-1}$) than RADI's in the top 15 cm, and higher in the deeper part of the sediment column.
- [section 3.2, last paragraph, added text]: The TAlk and ΣCO2 porewater profiles predicted by a RADI simulation using 4.5-order calcite dissolution kinetics are slightly lower (< ~40 µmol

kg$^{-1}$) than those using the new calcite dissolution kinetics scheme, and but the predicted calcite concentrations are slightly higher (< ~2%).

- [section 3.4, replaced last paragraph by]: In addition, we note than the choice of calcite dissolution kinetics implemented in RADI does not seem to have a large impact on TAlk and $\Sigma CO_2$ porewater profiles nor on the predicted calcite concentrations. RADI's step-edge retreat dissolution regime and its low reaction order induce calcite dissolution rates near equilibrium that are orders of magnitude higher than what is predicted in a high-order rate law (Fig. 2), but if the rate constant of a high-order rate law is tuned so that it overlaps the homogeneous dissolution rate law far from equilibrium, see Fig. 2, differences are limited. Thus, we conclude that using a 4.5-order rate law with a 10% day$^{-1}$ rate constant or using the new, mechanistic calcite dissolution rate scheme implemented in RADI should lead to similar predictions.

Specific comments:

L26: I thought we agreed that the model is not necessarily ready for ocean acidification events.

The model is not appropriate for long term acidification events that cause large chemical erosion, but should be appropriate to accurately predict the effects of ocean acidification on the short run, e.g., on dissolved fluxes across the sediment-water interface. We have added the adjective long-duration in L.810 to articulate this. We kept L.26 as it is because events are generally considered of short duration.

L340, L385: Is it possible that different biogenic or differently dissolved/aged CaCO3 could have different saturation states at which the rate law changes (currently around 0.8), assuming that the rate law close to equilibrium is controlled by density of steps/kinks that can vary with the extent of the reaction and/or between biological species?

This is possible, but Subhas et al. (2018, Marine Chemistry) showed that the switch between both regimes occurs at a $\Omega_{ca}$ that is very similar whether foraminifera, coccoliths or inorganic calcite are being dissolved. So based on the available data, this has not been observed so far.

Sections 3.1 and 3.2: Why the new rate law predicts higher preservation of CaCO3, given the higher dissolution rate at > 0.8 saturation ratio and the bottom water saturation ratio of 0.88 and 0.85 at these sites? Or is the bottom water at these sites still far from equilibrium so that traditional rate law predicts faster rate than the updated law? To make comparison clearer I think the authors should modify one of the rate laws so that rates of both models are the same/similar at far-from-equilibrium(please also see general comments).

This is comment is not relevant anymore, as we have followed the reviewer's suggestion of changing the rate constant of the high-order law, and the updated CaCO3 concentrations are now similar.

L677-L680: These lines could be incorrect. The bottom water chemistry with the saturation ratios of 0.88, 0.85 and 0.78 indicates that the dissolution rate could be higher with the traditional rate law (Fig. 2; please also see the comment above). In order to discuss the difference between the two models with the two rate laws with excluding any artificial difference caused by tuning, the authors have to normalize the two tunable rate laws at far from equilibrium region assuming that both are controlled by the same mechanisms at far from equilibrium (please also see general comments).

We have changed the rate constant of the high-order law, and updated this part.

L686: It would be useful to the reader if the authors mention the limitation of the model to transient application here, probably with an explanation on why the authors chose such specific application examples and not e.g., ocean acidification where we expect more drastic change of sedimentary CaCO3 system. This could be mentioned earlier as well, e.g., Section 1 or 2.

We have already mentioned that in the section entitled "additional applications", we believe that this is sufficient to inform readers on that specific limitation of the model.

**Reviewer #2**

Concerns about modeling alkalinity and DIC (versus the individual species that make them up) have largely been addressed. I still have some concern that they are only partially correct in their assumption that this only a problem at low pH. The other issue to consider is that in anoxic coastal sediments other species (e.g., dissolved sulfide, ammonium) may also contribute to the measured alkalinity, and this can only be addressed by modeling individual chemical species and then determining alkalinity after the fact by summing the relevant chemical components. I'm not sure how to address this here since this is not a problem in this work, and only may be a problem in future work. One possible suggestion is to add a sentence like this to line 821,

"Other chemical species (e.g., dissolved sulfide, ammonium) that also contribute to the measured pore water alkalinity may also invalidate this assumption."

(or words to this affect). However, I leave that decision to the editor.

Other than that, this manuscript looks ready for publication.

David Burdige

We thank David Burdige for this comment, and we have added the suggested sentence to line 821.